# Weaving in the Clouds: Achieving Synergistic Collaboration among LLM Agents via Federated Learning

**Jiaxing Zhao** [1]  **Hongbin Xie** [2]  **Yuzhen Lei** [1]  **Xuan Song** [1]  **Zhuoran Shi** [3]  **Lianxin Li** [3]  **Shuangxue Liu** [1]  **Haoran Zhang** [2]

## Abstract

Multi-Agent Systems (MAS) powered by Large Language Models (LLMs) have become a promising paradigm for solving workflow-structured tasks through expert collaboration. However, the data required for such collaboration are often distributed across organizations and cannot be centrally pooled due to privacy, intellectual property, and compliance constraints. Federated Learning preserves data locality, but existing federated paradigms typically treat clients as independent and fail to model workflow dependencies that are crucial for coherent multi-stage collaboration. We introduce **FedWave**, a federated and workflow-aware framework that enables LLM-based experts to solve sequential tasks under data-locality constraints. FedWave combines a *Value Chain Layer* to model inter-stage dependencies, federated LoRA adaptation to preserve role-specific expertise, a shared *Mixture-of-Experts (MoE)* router for input-conditioned expert fusion, and a *Direct Preference Optimization (DPO)* stage based on router-induced preferences. Experiments across workflow datasets and LLM backbones show that FedWave outperforms federated baselines and remains competitive with centralized multi-agent systems while preserving data locality.

## 1. Introduction

Large Language Models (LLMs) have demonstrated remarkable capabilities across a wide range of tasks, enabling increasingly complex real-world applications (Webb et al.,

2023; Ouyang et al., 2022; Achiam et al., 2023; Yang et al., 2024). A prominent class of these applications is *workflow-structured* problem solving, where the target solution must be produced through multiple *stages* with directed dependencies. Such workflows exhibit two defining properties: *stage specialization*, where each stage requires distinct expertise and data, and *dependency constraints*, where upstream outputs impose requirements on downstream decisions, so coherence depends on respecting the stage-to-stage order.

Multi-Agent Systems (MAS) built on LLMs have become a natural approach for workflow-structured tasks by assigning specialized roles to agents and coordinating them through a predefined or learned procedure (Zhao et al., 2024; Qian et al., 2025; Li et al., 2024). In domains such as business planning (Zhao et al., 2026), financial analysis (Yang et al., 2025), and medical diagnostics (Tang et al., 2024), MAS can model the workflow explicitly and reduce complex objectives into interdependent sub-decisions. However, many effective MAS pipelines rely on *centralized access* to large volumes of interaction traces, expert annotations, and domain documents (Cemri et al., 2025). This assumption is increasingly brittle because high-quality public data is becoming scarce (Villalobos et al., 2024), while valuable data and expertise are distributed across organizations and remain siloed due to privacy, IP, and compliance constraints (Ye et al., 2024; Fan et al., 2023). As a result, the key question is not only how to coordinate experts, but how to do so when data must remain local to each role.

Federated Learning (FL) offers a principled way to train models across parties without exchanging raw data (McMahan et al., 2017). By keeping data local and aggregating model updates, FL enables privacy-preserving collaboration and makes cross-organization training feasible. Yet most mainstream FL frameworks (Kuang et al., 2024; Yao et al., 2025) are not designed for workflow-structured settings: they typically treat clients as independent contributors and focus on mitigating statistical heterogeneity, while leaving *workflow dependencies* unmodeled. In practice, clients in a workflow are coupled by directed constraints, where upstream stages generate intermediate artifacts that downstream stages must follow. When aggregation ignores

---

[1]School of Artificial Intelligence, Jilin University [2]School of Urban Planning and Design, Peking University [3]Department of Computer Science and Engineering, Southern University of Science and Technology. Correspondence to: Xuan Song <songxuan@jlu.edu.cn>, Haoran Zhang <h.zhang@pku.edu.cn>.

*Proceedings of the 43rd International Conference on Machine Learning*, Seoul, South Korea. PMLR 306, 2026. Copyright 2026 by the author(s).

these couplings, updates from different stages can be incompatible, reducing coherence and limiting performance on multi-stage collaborative tasks, especially in non-IID and heterogeneous regimes (Li et al., 2020; Karimireddy et al., 2020; Hsu et al., 2019).

To address these challenges, we propose **FedWave**, a novel framework for federated multi-agent collaboration. Fed-Wave empowers LLM-based experts to collaboratively solve workflow-structured sequential tasks under strict privacy constraints, while producing a *single deployed model* for efficient inference. Its core mechanisms are as follows. First, at the client level, we combine FL with Parameter-Efficient Fine-Tuning (PEFT) via LoRA (Hu et al., 2022): each expert performs local LoRA-based adaptation on a shared backbone, substantially reducing trainable parameters, communication cost, and potential privacy leakage compared with full-model updates. Second, on the server, we introduce a *Mixture-of-Experts (MoE) router* (Shazeer et al., 2017; Fedus et al., 2022) to enable *input-conditioned, task-aware* fusion of expert knowledge at inference time. Notably, FedWave aggregates the shared router through standard FL while keeping expert-specific modules role-aware; the router replaces the implicit "uniform contribution" assumption with dynamic expert weighting when generating outputs. Finally, we apply a Direct Preference Optimization (DPO) stage (Rafailov et al., 2023) to align collaborative outcomes using *router-induced collaborative preferences*, improving coherence without sacrificing stage specialization. Extensive experiments demonstrate that FedWave outperforms strong FL baselines and is competitive with centralized multi-agent systems under realistic privacy constraints. The main contributions can be summarized as follows:

- **Workflow-aware Federated Collaboration:** We introduce **FedWave**, a framework enabling LLM agents to solve complex, workflow-structured sequential tasks across data silos. The proposed Value Chain Layer models inter-stage dependencies to support structured collaboration under privacy and data-locality constraints.

- **Dynamic, Task-Aware Expert Fusion:** We design a server-side MoE routing mechanism that performs input-conditioned expert fusion at inference time. Fed-Wave preserves FL-style training while moving beyond static averaging when generating collaborative outputs.

- **Preference Alignment for Collaborative Outputs:** We incorporate a DPO stage based on *router-induced collaborative preferences* to improve the coherence and quality of collaboration. Extensive experiments across multiple domains and backbones validate Fed-Wave's effectiveness against federated baselines and its competitiveness with centralized MAS baselines.

## 2. Related Work

### 2.1. Federated Learning

Federated Learning (FL) is a privacy-preserving paradigm for collaborative training on decentralized data (Kairouz et al., 2021). While FedAvg (McMahan et al., 2017) is widely used, its performance often degrades under Non-IID data and heterogeneous systems, motivating substantial follow-up work. Representative approaches mitigate heterogeneity via regularization or control variates (e.g., FedProx (Li et al., 2020), SCAFFOLD (Karimireddy et al., 2020)), or improve server-side aggregation dynamics (e.g., FedNova (Wang et al., 2020), FedAvgM (Hsu et al., 2019)). However, most FL work targets statistical heterogeneity and treats clients as independent. In workflow settings, clients are dependency-coupled, and *FedWave* makes this structure explicit to enable coherent federated collaboration.

### 2.2. Multi-Agent Collaboration

The rise of Large Language Models (LLMs) has significantly advanced Multi-Agent Systems (MAS), establishing them as a powerful paradigm for solving complex problems via role specialization and coordination (Akata et al., 2023; Guo et al., 2024; Hao et al., 2025). Existing work explores diverse interaction patterns, including discussion and debate to improve reasoning (Du et al., 2024; Chen et al., 2024; Xiong et al., 2023), as well as hierarchical and workflow-style pipelines (Zhang et al., 2024; Zhao et al., 2024; 2026). However, many MAS pipelines assume centralized data or shared memory, which breaks in privacy-siloed settings. *FedWave* enables workflow-style coordination under federation while retaining a single coherent deployable model.

### 2.3. MoE Routing for Expert Fusion

Mixture-of-Experts (MoE) scales models via conditional computation, using a router to select experts per input (Jacobs et al., 1991; Fedus et al., 2022; Shazeer et al., 2017). Related ideas appear in federated personalization, where routing or client-specific components help handle heterogeneity (Fallah et al., 2020; Arivazhagan et al., 2019; T Dinh et al., 2020; Li et al., 2021). However, prior work rarely couples input-conditioned expert selection with workflow structure under standard FL aggregation. *FedWave* closes this gap by co-training a lightweight router and workflow-aware representations, then using routing for task-aware expert fusion at inference time.

## 3. Methods

### 3.1. Federated Fine-Tuning with the Value Chain Layer

The first phase of our framework fine-tunes a base LLM in a federated setting where each client, an expert agent (e.g.,

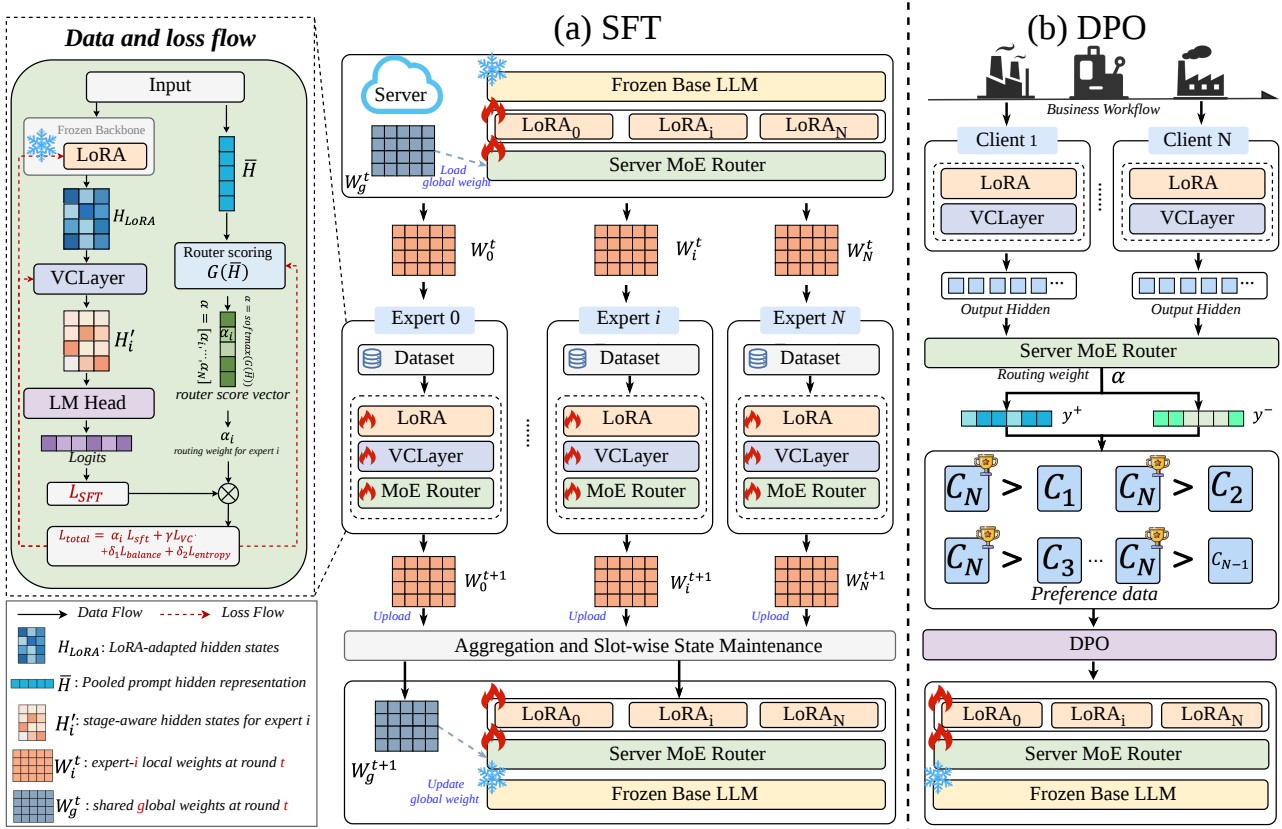

*Figure 1.* Overview of the FedWave framework. The left panel illustrates the local data and loss flow inside each expert client, where LoRA-adapted hidden states are processed by the VCLayer and the pooled prompt representation is scored by the MoE router to produce routing weights. Panel (a) shows federated SFT: the server distributes trainable states to expert clients, each client updates its LoRA, VCLayer, and router parameters on private workflow data, and the server aggregates the shared router while maintaining expert modules by role. Panel (b) shows DPO alignment: the trained router ranks candidate expert responses, constructs preference pairs from top-ranked and lower-ranked responses, and aligns the final model through DPO. Black arrows denote data flow, red dashed arrows denote loss flow, and the weight blocks indicate the transition from round-$t$ local/global parameters to the updated round-$(t+1)$ parameters.

design, manufacturing), holds a private dataset for their role. To model the sequential relationships between these experts, we introduce the Value Chain Layer (VCLayer). As shown in Figure 1 (a), each expert agent $i$ has a frozen base LLM, a trainable LoRA adapter ($W_i$), and our VCLayer. This lightweight, pluggable module processes the hidden states from the LoRA-adapted model, making them aware of the agent's role and position in the workflow.

Given the final hidden states $H^{\text{LoRA}} \in \mathbb{R}^{L \times d}$ from the backbone augmented with the role-specific LoRA adapter, the VCLayer applies a stage-specific transformation. For an agent at stage $i$, we use a specialized multi-head attention module, denoted as $\text{MHA}_i(\cdot)$:

$$H_i' = \text{MHA}_i\big(H^{\text{LoRA}}\big). \tag{1}$$

Specifically, we compute $[Q_i, K_i, V_i] = H^{\text{LoRA}} W_{qkv,i}$ (split along the last dimension), and apply $\text{MHA}_i(H) = \text{Attention}(Q_i, K_i, V_i) W_{o,i}$, where $W_{qkv,i} \in \mathbb{R}^{d \times 3d_{vc}}$ and $W_{o,i} \in \mathbb{R}^{d_{vc} \times d}$ are trainable, stage-specific projection ma-

trices. We denote by $W_{q,i}$ the query block of $W_{qkv,i}$ (i.e., the sub-matrix that produces $Q_i$). This allows each expert to focus on task-relevant aspects of the input. The resulting state $H_i'$ is then passed to the LM head to produce logits. A key innovation is the collaborative loss, $\mathcal{L}_{VC}$, which guides the VCLayer to learn the workflow structure. In practice, these structural terms are instantiated against the previous-round global snapshot of neighboring stages, and the consistency term is evaluated on a small set of role-agnostic anchor prompts shared across stages (without using any private labels). During local training for agent $i$, the total loss combines the standard Supervised Fine-Tuning (SFT) loss with our collaborative loss:

$$\mathcal{L}_{\text{SFT+VC}}^{(i)} = \mathcal{L}_{\text{SFT}}^{(i)} + \gamma \mathcal{L}_{VC}^{(i)}. \tag{2}$$

Here, $\gamma$ is a balancing hyperparameter. $\mathcal{L}_{\text{SFT}}$ is the conventional cross-entropy loss for next-token prediction. $\mathcal{L}_{VC}$ comprises three terms that enforce the value chain's rela-

tional structure:

$$\mathcal{L}_{VC} = \lambda_{\text{pos}}\mathcal{L}_{\text{pos}} + \lambda_{\text{cont}}\mathcal{L}_{\text{cont}} + \lambda_{\text{cons}}\mathcal{L}_{\text{cons}} \qquad (3)$$

where $\lambda_{\text{pos}}$, $\lambda_{\text{cont}}$, and $\lambda_{\text{cons}}$ are weighting coefficients.

**Positional Loss ($\mathcal{L}_{\textbf{pos}}$):** This loss enforces a geometric arrangement of experts in an embedding space, reflecting their sequential order. It is based on the similarity between learnable embeddings $\{e_0, ..., e_{N-1}\}$ for each stage:

$$\mathcal{L}_{\text{pos}} = \sum_{i=0}^{N-2} \left( \cos(e_i, e_{i+1}) - T_{\text{pos}}^{(i)} \right)^2 \qquad (4)$$

where $T_{\text{pos}}^{(i)}$ is a target cosine similarity, encouraging adjacent experts to be closer.

**Continuity Loss ($\mathcal{L}_{\textbf{cont}}$):** This loss promotes a smooth transition of knowledge by ensuring adjacent experts learn similar functions. It operates on the VCLayer's query projections (we use $W_{q,i}$ as a lightweight proxy for functional similarity):

$$\mathcal{L}_{\text{cont}} = \sum_{i=0}^{N-2} \left( \cos(\text{vec}(W_{q,i}), \text{vec}(W_{q,i+1})) - T_{\text{cont}} \right)^2 \quad (5)$$

where $\text{vec}(\cdot)$ is the vectorization operator and $T_{\text{cont}}$ is a high target similarity value, encouraging the attention mechanisms of consecutive stages to be functionally alike.

**Consistency Loss ($\mathcal{L}_{\textbf{cons}}$):** This loss ensures a coherent solution progression by aligning an expert's output representation with its predecessor's on the same anchor prompt $x \sim \mathcal{P}_{\text{anc}}$. For an agent at stage $i > 0$, using the previous-round snapshot for stage $i{-}1$:

$$\mathcal{L}_{\text{cons}}^{(i)} = \left( \cos\left( f_i(H_i^{\text{LoRA}}(x)), \ f_{i-1}(H_{i-1}^{\overline{\text{LoRA}}}(x)) \right) - T_{\text{cons}} \right)^2, \tag{6}$$

where $H_j^{\text{LoRA}}(x)$ denotes the final hidden states produced by the LoRA-augmented model of stage $j$ (for $j \in \{i, i{-}1\}$), and $f_j(H)$ is the mean-pooled VCLayer output of stage $j$ given $H$.

**General workflow graphs and scalability of VCLayer.** For notational simplicity, we describe the workflow as a linear chain of $N$ experts in the main text. More generally, practical workflows can be represented as a directed graph $\mathcal{G} = (\mathcal{V}, \mathcal{E})$, where each node $u \in \mathcal{V}$ is an expert and each edge $(u,v) \in \mathcal{E}$ encodes a dependency $u \to v$. In this setting, the VCLayer losses naturally generalize by

summing over edges:

$$\begin{aligned}
\mathcal{L}_{\text{pos}} &= \sum_{(u,v)\in\mathcal{E}} \ell_{\text{pos}}(e_u, e_v), \\
\mathcal{L}_{\text{cont}} &= \sum_{(u,v)\in\mathcal{E}} \ell_{\text{cont}}(W_{q,u}, W_{q,v}), \\
\mathcal{L}_{\text{cons}} &= \sum_{(u,v)\in\mathcal{E}} \ell_{\text{cons}}(f_u(H), f_v(H)).
\end{aligned} \qquad (7)$$

where $e_u$ is the stage embedding, $W_{q,u}$ the VCLayer query projection, and $f_u(H)$ the VCLayer output at expert $u$. Thus, VCLayer is a set of local constraints over the edges of a workflow graph and its cost scales with the number of dependencies $|\mathcal{E}|$ (not $\mathcal{O}(N^2)$ over all pairs), making it practical for workflows with many stages and sparse structure.

### 3.2. Inference-Time Expert Fusion with MoE Routing

To move beyond the implicit "uniform contribution" assumption in federated deployment, we introduce a trainable Mixture-of-Experts (MoE) router as an input-conditioned coordinator. The router assigns prompt-dependent weights to experts and enables task-aware fusion at inference time, while its shared parameters are aggregated through standard FL during training. As a shared global component, the router is co-trained with each expert's LoRA adapters and VCLayer in the federated SFT phase.

The MoE Router $G(\cdot)$ is implemented as a lightweight MLP. For any input, it first computes a prompt representation by mean-pooling the base LLM's hidden states, $\bar{H} = \frac{1}{L}\sum_{l=1}^{L} H_l$. It then produces logits $z \in \mathbb{R}^N$ over the $N$ experts, which are converted to routing weights by:

$$\alpha = \text{softmax}(G(\bar{H})), \qquad (8)$$

where $\alpha_i$ denotes the routing weight for expert $i$. During local SFT on client $i$, we use $\alpha_i$ as a differentiable gate to modulate the expert's training signal: the local SFT loss is weighted by $\alpha_i$, so that when the router assigns higher relevance to expert $i$ for a prompt, the corresponding gradient signal is strengthened. This couples router learning with expert adaptation via end-to-end backpropagation. Since each client corresponds to a workflow role, this training provides weak supervision that prompts from role $i$ should place higher mass on expert $i$, enabling the aggregated router to learn input-conditioned specialization rather than a constant selector.

We add standard MoE regularizers to stabilize routing (load balancing and entropy control). A **load balancing loss** $\mathcal{L}_{\text{balance}}$ encourages balanced utilization across experts in a batch to prevent specialization collapse. An **entropy-based confidence loss** $\mathcal{L}_{\text{entropy}}$ discourages overly uncertain routing and promotes sparse, confident assignments. The

*Table 1.* Performance comparison of FedWave and baselines across three workflow datasets and three different backbone models. The best scores for each metric are highlighted in **bold**. ↑ / ↓ denotes the absolute change compared to *FedAvg* within the same backbone.

| Baselines | Automotive | | | E-commerce | | | Pharmaceutical | | |
|---|---|---|---|---|---|---|---|---|---|
| | **BS-F** | **Meteor** | **Rouge-L** | **BS-F** | **Meteor** | **Rouge-L** | **BS-F** | **Meteor** | **Rouge-L** |
| *Qwen2-7B* | | | | | | | | | |
| FedAvg | 71.11 | 23.35 | 22.18 | 79.17 | 43.29 | 38.68 | 77.12 | 26.76 | 28.54 |
| FedAvgM | $70.66^{\downarrow0.45}$ | $22.02^{\downarrow1.33}$ | $21.74^{\downarrow0.44}$ | $79.51^{\uparrow0.34}$ | $43.09^{\downarrow0.20}$ | $39.23^{\uparrow0.55}$ | $76.87^{\downarrow0.25}$ | $25.57^{\downarrow1.19}$ | $27.51^{\downarrow1.03}$ |
| FedProx | $71.36^{\uparrow0.25}$ | $23.58^{\uparrow0.23}$ | $22.53^{\uparrow0.35}$ | $79.23^{\uparrow0.06}$ | $43.53^{\uparrow0.24}$ | $38.84^{\uparrow0.16}$ | $77.18^{\uparrow0.06}$ | $27.08^{\uparrow0.32}$ | $28.65^{\uparrow0.11}$ |
| FedAdam | $71.39^{\uparrow0.28}$ | $23.74^{\uparrow0.39}$ | $21.74^{\downarrow0.44}$ | $78.78^{\downarrow0.39}$ | $41.01^{\downarrow2.28}$ | $37.07^{\downarrow1.61}$ | $76.41^{\downarrow0.71}$ | $26.07^{\downarrow0.69}$ | $27.29^{\downarrow1.25}$ |
| FedYogi | $71.49^{\uparrow0.38}$ | $24.18^{\uparrow0.83}$ | $22.48^{\uparrow0.30}$ | $78.54^{\downarrow0.63}$ | $40.31^{\downarrow2.98}$ | $36.81^{\downarrow1.87}$ | $76.00^{\downarrow1.12}$ | $24.59^{\downarrow2.17}$ | $26.11^{\downarrow2.43}$ |
| Scaffold | $71.26^{\uparrow0.15}$ | $23.53^{\uparrow0.18}$ | $22.49^{\uparrow0.31}$ | $79.04^{\downarrow0.13}$ | $42.32^{\downarrow0.97}$ | $38.50^{\downarrow0.18}$ | $77.06^{\downarrow0.06}$ | $27.09^{\uparrow0.33}$ | $28.68^{\uparrow0.14}$ |
| **FedWave** | $\mathbf{71.90}^{\uparrow0.79}$ | $\mathbf{40.35}^{\uparrow17.00}$ | $\mathbf{23.46}^{\uparrow1.28}$ | $\mathbf{80.17}^{\uparrow1.00}$ | $\mathbf{48.28}^{\uparrow4.99}$ | $\mathbf{39.63}^{\uparrow0.95}$ | $\mathbf{78.34}^{\uparrow1.22}$ | $\mathbf{29.82}^{\uparrow3.06}$ | $\mathbf{31.89}^{\uparrow3.35}$ |
| *Llama2-7B* | | | | | | | | | |
| FedAvg | 69.43 | 17.41 | 20.98 | 72.96 | 16.45 | 22.66 | 70.61 | 10.10 | 14.98 |
| FedAvgM | $65.62^{\downarrow3.81}$ | $14.32^{\downarrow3.09}$ | $16.65^{\downarrow4.33}$ | $73.11^{\uparrow0.15}$ | $18.13^{\uparrow1.68}$ | $23.73^{\uparrow1.07}$ | $70.95^{\uparrow0.34}$ | $11.24^{\uparrow1.14}$ | $15.15^{\uparrow0.17}$ |
| FedProx | $69.26^{\downarrow0.17}$ | $17.08^{\downarrow0.33}$ | $20.74^{\downarrow0.24}$ | $72.82^{\downarrow0.14}$ | $16.30^{\downarrow0.15}$ | $22.52^{\downarrow0.14}$ | $70.92^{\uparrow0.31}$ | $10.27^{\uparrow0.17}$ | $15.06^{\uparrow0.08}$ |
| FedAdam | $68.90^{\downarrow0.53}$ | $17.03^{\downarrow0.38}$ | $19.55^{\downarrow1.43}$ | $70.83^{\downarrow2.13}$ | $13.49^{\downarrow2.96}$ | $19.88^{\downarrow2.78}$ | $68.79^{\downarrow1.82}$ | $8.75^{\downarrow1.35}$ | $13.34^{\downarrow1.64}$ |
| FedYogi | $69.12^{\downarrow0.31}$ | $17.20^{\downarrow0.21}$ | $19.99^{\downarrow0.99}$ | $70.42^{\downarrow2.54}$ | $13.15^{\downarrow3.30}$ | $19.40^{\downarrow3.26}$ | $68.36^{\downarrow2.25}$ | $8.31^{\downarrow1.79}$ | $12.94^{\downarrow2.04}$ |
| Scaffold | $69.25^{\downarrow0.18}$ | $16.78^{\downarrow0.63}$ | $20.64^{\downarrow0.34}$ | $72.95^{\downarrow0.01}$ | $16.62^{\uparrow0.17}$ | $22.72^{\uparrow0.06}$ | $70.97^{\uparrow0.36}$ | $10.39^{\uparrow0.29}$ | $15.25^{\uparrow0.27}$ |
| **FedWave** | $\mathbf{70.08}^{\uparrow0.65}$ | $\mathbf{18.55}^{\uparrow1.14}$ | $\mathbf{22.12}^{\uparrow1.14}$ | $\mathbf{74.95}^{\uparrow1.99}$ | $\mathbf{20.80}^{\uparrow4.35}$ | $\mathbf{26.30}^{\uparrow3.64}$ | $\mathbf{74.78}^{\uparrow4.17}$ | $\mathbf{13.38}^{\uparrow3.28}$ | $\mathbf{19.73}^{\uparrow4.75}$ |
| *Llama3-8B* | | | | | | | | | |
| FedAvg | 58.62 | 10.73 | 9.55 | 68.81 | 26.25 | 25.29 | 61.34 | 14.17 | 14.49 |
| FedAvgM | $56.03^{\downarrow2.59}$ | $7.33^{\downarrow3.40}$ | $6.76^{\downarrow2.79}$ | $64.61^{\downarrow4.20}$ | $24.05^{\downarrow2.20}$ | $21.19^{\downarrow4.10}$ | $61.84^{\uparrow0.50}$ | $14.24^{\uparrow0.07}$ | $14.38^{\downarrow0.11}$ |
| FedProx | $58.44^{\downarrow0.18}$ | $10.50^{\downarrow0.23}$ | $9.27^{\downarrow0.28}$ | $68.52^{\downarrow0.29}$ | $27.16^{\uparrow0.91}$ | $25.66^{\uparrow0.37}$ | $61.79^{\uparrow0.45}$ | $14.50^{\uparrow0.33}$ | $15.06^{\uparrow0.57}$ |
| FedAdam | $51.47^{\downarrow7.15}$ | $2.67^{\downarrow8.06}$ | $2.25^{\downarrow7.30}$ | $57.22^{\downarrow11.59}$ | $18.28^{\downarrow7.97}$ | $14.65^{\downarrow10.64}$ | $58.08^{\downarrow3.26}$ | $12.16^{\downarrow2.01}$ | $11.49^{\downarrow3.00}$ |
| FedYogi | $52.78^{\downarrow5.84}$ | $4.17^{\downarrow6.56}$ | $3.66^{\downarrow5.89}$ | $57.37^{\downarrow11.44}$ | $18.16^{\downarrow8.09}$ | $14.77^{\downarrow10.52}$ | $58.20^{\downarrow3.14}$ | $11.68^{\downarrow2.49}$ | $11.05^{\downarrow3.44}$ |
| Scaffold | $59.15^{\uparrow0.53}$ | $11.31^{\uparrow0.58}$ | $10.21^{\uparrow0.66}$ | $68.12^{\downarrow0.69}$ | $25.65^{\downarrow0.60}$ | $24.60^{\downarrow0.69}$ | $61.62^{\uparrow0.28}$ | $14.85^{\uparrow0.68}$ | $14.97^{\uparrow0.48}$ |
| **FedWave** | $\mathbf{70.27}^{\uparrow11.65}$ | $\mathbf{24.83}^{\uparrow14.10}$ | $\mathbf{17.47}^{\uparrow7.92}$ | $\mathbf{79.66}^{\uparrow10.85}$ | $\mathbf{44.72}^{\uparrow18.47}$ | $\mathbf{38.33}^{\uparrow13.04}$ | $\mathbf{77.95}^{\uparrow16.61}$ | $\mathbf{24.02}^{\uparrow9.85}$ | $\mathbf{28.40}^{\uparrow13.91}$ |

total local loss on client $i$ is:

$$\mathcal{L}_{\text{total}}^{(i)} = \alpha_i \mathcal{L}_{\text{SFT}}^{(i)} + \gamma \mathcal{L}_{VC}^{(i)} + \delta_1 \mathcal{L}_{\text{balance}} + \delta_2 \mathcal{L}_{\text{entropy}}, \quad (9)$$

where $\delta_1$ and $\delta_2$ are hyperparameters. This composite objective enables joint training of the LoRA adapters, VCLayer, and the MoE router.

After each local training round, we aggregate the shared router parameters on the server:

$$W_{\text{router}}^{t+1} = \sum_{i \in S_t} \frac{n_i}{n} W_{\text{router},i}^{t+1}. \quad (10)$$

LoRA adapters and VCLayers follow the same round structure but remain associated with their corresponding workflow roles. At inference time, the learned routing weights $\alpha$ are used to perform input-conditioned expert fusion (optionally with top-$k$ routing for efficiency), enabling dynamic, task-aware utilization of heterogeneous expertise.

### 3.3. Preference Alignment with DPO

After the federated SFT phase, the aggregated model has acquired stage-specialized knowledge and a foundational understanding of workflow structure. The final phase (Figure 1(b)) further improves output coherence by aligning the model with *collaborative* preferences using DPO (Rafailov et al., 2023). Importantly, the preference signal here is not collected from humans; instead, we construct *router-induced collaborative preferences* by leveraging the co-trained MoE router to create comparable alternative outcomes under the same prompt. Concretely, we use the final SFT model $\pi_{\text{SFT}}$ to automatically build a preference dataset $\mathcal{D}_{\text{pref}} = \{(x, y^+, y^-)\}$. For each prompt $x$, we:

1. Compute routing weights $\alpha$ with the MoE router inside $\pi_{\text{SFT}}$, and rank the $N$ experts by relevance to $x$.

2. Select the top-ranked expert $c^+ = \arg\max_i \alpha_i$ to generate a *preferred* response $y^+$.

3. Select a lower-ranked expert $c^-$ to generate an alternative, *dispreferred* response $y^-$. We sample $c^-$ uniformly from the non-top experts (excluding the router top-1, or excluding top-$k$ when using top-$k$ routing) to form diverse but comparable negatives.

4. Add $(x, y^+, y^-)$ to $\mathcal{D}_{\text{pref}}$.

*Table 2.* Performance comparison of FedWave against centralized multi-agent baselines on the Qwen2-7B backbone. FedWave operates in a decentralized, privacy-preserving setting, while the baselines have access to the full, centralized dataset. The best scores for each metric are highlighted in **bold**.

| Dataset | Baselines | BS-F | GLEU | BLEU-4 | ROUGE-1 | ROUGE-2 | ROUGE-L |
|---|---|---|---|---|---|---|---|
| Automotive | PMC (Zhang et al., 2025) | 65.81 | 13.25 | 5.24 | 26.44 | 4.51 | 18.37 |
| | MedAgents (Tang et al., 2024) | 64.94 | 12.47 | 4.81 | 23.68 | 4.07 | 18.34 |
| | Debate(long) (Du et al., 2024) | 65.56 | 12.92 | 6.49 | 25.49 | 5.91 | 18.03 |
| | Debate(short) (Du et al., 2024) | 65.32 | 12.66 | 6.30 | 25.39 | 6.02 | 17.81 |
| | CoA (Zhang et al., 2024) | 70.77 | **22.30** | 14.41 | 33.99 | 11.15 | **24.01** |
| | **FedWave (Ours)** | **71.90** | 20.42 | **15.44** | **35.11** | **12.19** | 23.46 |
| E-commerce | PMC (Zhang et al., 2025) | 70.88 | 24.01 | 19.73 | 33.47 | 11.45 | 29.91 |
| | MedAgents (Tang et al., 2024) | 69.57 | 17.48 | 12.60 | 33.62 | 10.77 | 28.25 |
| | Debate(long) (Du et al., 2024) | 72.89 | 20.81 | 15.46 | 36.43 | 13.12 | 28.04 |
| | Debate(short) (Du et al., 2024) | 72.93 | 20.85 | 15.56 | 37.11 | 13.51 | 28.23 |
| | CoA (Zhang et al., 2024) | 79.20 | 38.08 | 34.23 | 50.02 | 25.35 | 39.02 |
| | **FedWave (Ours)** | **80.17** | **42.60** | **39.56** | **51.04** | **26.22** | **39.63** |
| Pharmaceutical | PMC (Zhang et al., 2025) | 68.55 | 17.34 | 11.67 | 28.16 | 7.52 | 25.93 |
| | MedAgents (Tang et al., 2024) | 65.96 | 10.60 | 4.46 | 25.91 | 6.17 | 21.13 |
| | Debate(long) (Du et al., 2024) | 61.14 | 9.58 | 5.33 | 16.18 | 4.69 | 12.23 |
| | Debate(short) (Du et al., 2024) | 70.57 | 15.78 | 9.11 | 31.76 | 9.60 | 23.03 |
| | CoA (Zhang et al., 2024) | 77.26 | **29.92** | 18.18 | 42.38 | 19.93 | **33.70** |
| | **FedWave (Ours)** | **78.34** | 26.27 | **19.22** | **43.24** | **20.00** | 31.89 |

This automated construction yields large-scale preference pairs that favor outputs produced under the router's most contextually appropriate specialization, serving as a lightweight signal for improving collaboration quality.

We then fine-tune the policy $\pi_\theta$ on $\mathcal{D}_{\text{pref}}$ with DPO. The policy is initialized from $\pi_{\text{SFT}}$, and a frozen copy of $\pi_{\text{SFT}}$ is used as the reference model $\pi_{\text{ref}}$, which regularizes updates to stay close to the SFT solution. The DPO loss is:

$$\mathcal{L}_{\text{DPO}}(\pi_\theta; \pi_{\text{ref}}) = -\log \sigma \left( \beta \log \frac{\pi_\theta(y^+|x)\, \pi_{\text{ref}}(y^-|x)}{\pi_\theta(y^-|x)\, \pi_{\text{ref}}(y^+|x)} \right). \tag{11}$$

where $\sigma$ is the logistic function and $\beta$ controls the preference strength. By minimizing this loss, FedWave performs a *regularized* alignment step that improves output coherence while preserving the stage specialization learned during federated SFT. The resulting model constitutes the final FedWave policy for efficient, high-quality workflow collaboration under privacy constraints.

## 4. Experiments

Our experiments evaluate FedWave to address four questions: **(1)** How does its collaborative performance on sequential tasks compare to standard federated learning? **(2)** How does the privacy-preserving FedWave perform against a centralized multi-agent system with full data access? **(3)** Which design elements are most critical to its success? **(4)** How does FedWave scale and remain robust across federated protocols and deployment conditions?

### 4.1. Experimental Setup

**Datasets and evaluation metrics.** We evaluate our framework mainly on the MSCoRe benchmark (Lei et al., 2025), which is specifically designed for multi-stage collaborative reasoning. It provides three challenging datasets with complex, sequential tasks representing distinct business workflows: **E-commerce**, **Pharmaceutical**, and **Automotive**. To further examine robustness across different reasoning scenarios, we additionally evaluate FedWave on two public benchmarks, **MuSiQue** (Trivedi et al., 2022) and **FinQA** (Chen et al., 2021). MuSiQue evaluates multi-hop question answering, while FinQA focuses on financial numerical reasoning; for FinQA, we convert the original instances into a final-answer QA format and apply answer normalization for evaluation. For the MSCoRe workflow datasets, we employ a diverse suite of metrics beyond simple lexical overlap, including ROUGE-1/2/L (Lin, 2004) for lexical content, BLEU-4 (Papineni et al., 2002) and GLEU (Wu et al., 2016), and METEOR (Banerjee & Lavie, 2005) and BERTScore (Zhang* et al., 2020) to evaluate semantic fidelity and contextual relevance. For MuSiQue and processed FinQA, we report EM and token-level F1 following standard question-answering evaluation protocols.

**Baselines.** To validate our framework's effectiveness, we compare it against two baseline categories. For federated learning, we adapt widely-recognized algorithms: the foundational FedAvg (McMahan et al., 2017); FedProx (Li et al., 2020) with its proximal term to mitigate heterogeneity; FedAvgM (Hsu et al., 2019), which adds server-side momentum; SCAFFOLD (Karimireddy et al., 2020) for client-

Table 3. Results on additional public benchmarks beyond the main MSCoRe workflow datasets. FedWave is compared with federated baselines under the same backbone and training protocol.

| Benchmark | Method | EM | F1 |
|---|---|---|---|
| MuSiQue | FedAdam | 3.39 | 10.93 |
| | FedAvg | 3.10 | 11.21 |
| | FedProx | 3.23 | 11.38 |
| | FedWave | **7.20** | **19.27** |
| FinQA | FedAdam | 0.50 | 4.13 |
| | FedAvg | 0.50 | **5.58** |
| | FedProx | 0.50 | 4.42 |
| | FedWave | **3.00** | 3.33 |

Table 4. Component ablations on the Automotive workflow using the Llama3-8B backbone. We report METEOR as the primary metric.

| Variant | VCLayer | Router | DPO | Meteor |
|---|---|---|---|---|
| FedAvg-LoRA (baseline) | ✗ | ✗ | ✗ | 10.73 |
| *- Single Component -* | | | | |
| + VCLayer only | ✓ | ✗ | ✗ | 17.93 |
| + Router only | ✗ | ✓ | ✗ | 17.52 |
| + DPO only | ✗ | ✗ | ✓ | 21.04 |
| *- Two Components -* | | | | |
| + VCLayer + Router | ✓ | ✓ | ✗ | 18.40 |
| + VCLayer + DPO | ✓ | ✗ | ✓ | 23.65 |
| + Router + DPO | ✗ | ✓ | ✓ | 23.95 |
| **FedWave (full)** | ✓ | ✓ | ✓ | **24.83** |

drift correction; and the adaptive optimizers FedAdam and FedYogi (Reddi et al., 2021). For multi-agent systems, we benchmark against centralized methods that operate on an aggregated dataset: the hierarchical PMC (Zhang et al., 2025), discussion-based MedAgents (Tang et al., 2024), chain-based CoA (Zhang et al., 2024), and Debate (Du et al., 2024), which enables collaboration through argumentative discourse by modulating agent confidence through varying debate durations. For all federated baselines, we match FedWave in backbone, LoRA configuration, local update budget, and communication rounds, and only replace our workflow-aware components with the corresponding optimizer. The aggregated global model is then directly used to answer end-to-end workflow prompts under the same prompt template and decoding setup.

**Details.** We use 7B/8B-scale LLMs (Qwen2-7B, Llama2-7B (Touvron et al., 2023), Llama3-8B (Grattafiori et al., 2024)) as base backbones. For generation efficiency, inference is run with 8-bit quantization; training and communication use 16-bit parameters. FedWave uses 4 expert clients, with full participation in each communication round unless stated otherwise. In the federated SFT phase, we run 20 communication rounds; in each round, every client trains locally for 10 steps with AdamW (batch size 4) using a cosine schedule from $5 \times 10^{-5}$ to $1 \times 10^{-6}$. The maximum sequence length is 2048. We use LoRA with rank 32 and $\alpha = 64$. The MoE router uses top-$k$ routing with $k = 2$. VCLayer weights are set to $\lambda_{pos} = 0.1$, $\lambda_{cont} = 0.1$, $\lambda_{cons} = 0.2$, and the VC balance coefficient is $\gamma = 0.5$. For preference alignment, we fine-tune with DPO for 10 epochs with learning rate $1 \times 10^{-5}$ and $\beta = 0.1$. All experiments are conducted on NVIDIA A40 GPUs. We use the Alpaca (Taori et al., 2023) template to format.

### 4.2. FedWave vs. Federated Baselines

We present the main experimental results in Table 1, comparing FedWave against six federated learning baselines across three business workflow datasets and three LLM backbones. FedWave achieves the best results across the re-

ported MSCoRe settings, with especially large gains on ME-TEOR and BERTScore-F, suggesting that workflow-aware modeling mainly improves semantic coherence and contextual relevance. For instance, on the Automotive dataset with the Qwen2-7B backbone, FedWave achieves a METEOR score of 40.35, substantially improving upon the strongest federated baseline score of 24.18. These results indicate that explicitly modeling sequential dependencies and using input-conditioned expert fusion are beneficial under federated workflow constraints. The gains are also observed across different backbone families, suggesting that FedWave is not tied to a single model architecture.

To further examine the transferability of FedWave across different reasoning scenarios, we additionally evaluate it on two public benchmarks, MuSiQue and FinQA. As shown in Table 3, FedWave transfers strongly to MuSiQue, substantially improving both EM and F1 over federated baselines. On FinQA, FedWave also achieves a clear improvement in EM, although its F1 score remains lower than FedAvg. These results support FedWave's transferability while also showing that numerical reasoning remains challenging.

### 4.3. FedWave vs. Centralized Multi-Agent Baselines

We further benchmark FedWave against several *centralized* multi-agent baselines in Table 2. The two settings differ fundamentally: the centralized multi-agent baselines assume full visibility of all data and shared interaction context, which we include as an *upper-bound simulation* of idealized cross-role coordination. In contrast, FedWave is trained and deployed under federated constraints, where data remain siloed and only model updates are exchanged. Under this stricter setting, FedWave remains competitive with strong centralized MAS baselines. On E-commerce, FedWave surpasses all centralized baselines across all reported metrics. On Automotive and Pharmaceutical, FedWave obtains the best BERTScore-F, BLEU-4, ROUGE-1, and ROUGE-2 scores. These results show that FedWave can recover much

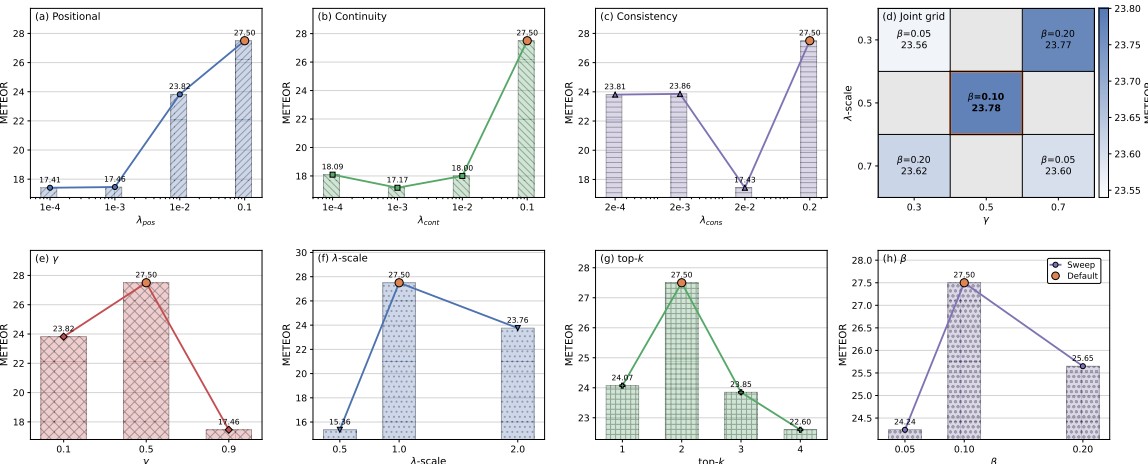

*Figure 2.* Hyperparameter sensitivity of FedWave on Automotive (Llama3-8B). We report METEOR under one-factor sweeps for VCLayer loss weights $\lambda_{\text{pos}}$, $\lambda_{\text{cont}}$, $\lambda_{\text{cons}}$, and core hyperparameters $\gamma$ (SFT–VC trade-off), $\lambda$-scale (shared scaling of VCLayer weights), router top-$k$, and DPO strength $\beta$. The orange marker indicates the default configuration used in the main experiments. Panel (d) shows a small joint grid over $\lambda$-scale, $\gamma$, and $\beta$, where performance varies within a narrow band around the default setting, suggesting that FedWave remains stable under coupled hyperparameter changes.

*Table 5.* Fine-grained ablation of VCL losses and router regularizers on the Automotive workflow using the Llama3-8B backbone. We report METEOR as the primary metric.

| Variant | METEOR |
|---|---|
| Full FedWave | **24.83** |
| w/o $\mathcal{L}_{pos}$ | $24.41^{\downarrow 0.42}$ |
| w/o $\mathcal{L}_{cont}$ | $24.30^{\downarrow 0.53}$ |
| w/o $\mathcal{L}_{cons}$ | $24.28^{\downarrow 0.55}$ |
| w/o $\mathcal{L}_{balance}$ | $24.32^{\downarrow 0.51}$ |
| w/o $\mathcal{L}_{entropy}$ | $24.57^{\downarrow 0.26}$ |
| w/o $\mathcal{L}_{balance} + \mathcal{L}_{entropy}$ | $24.17^{\downarrow 0.66}$ |

*Table 6.* Client scaling under fixed workflow roles on the Automotive workflow using the Llama3-8B backbone. The number of workflow roles is fixed to 4, while each role is split into multiple federated clients. For the multi-client settings, we keep 4 participating clients per round.

| Setting | Total clients | METEOR |
|---|---|---|
| 1 client per role | 4 | 24.83 |
| 2 clients per role | 8 | **25.31** |
| 4 clients per role | 16 | 25.07 |
| 8 clients per role | 32 | 25.28 |

of the benefit of workflow-style collaboration without relaxing data locality.

### 4.4. Key Design Factors and Hyperparameter Influence

**Ablation Study.** Table 4 disentangles the contribution of each major component in FedWave. Starting from the FedAvg-LoRA baseline (10.73 METEOR), enabling only *VCLayer*, *MoE Router*, or *DPO* improves performance to 17.93, 17.52, and 21.04, respectively, showing that workflow-aware modeling, input-conditioned expert utilization, and preference alignment are all beneficial. The components are also **complementary rather than redundant**: combining *VCLayer* and *Router* reaches 18.40 without DPO, while adding DPO to either structural module further improves performance to 23.65 / 23.95. The full FedWave model achieves the best result of 24.83 METEOR, bringing an additional +3.79 over the DPO-only configuration and indicating that DPO aligns the final collaborative behavior,

whereas VCLayer and Router determine *who contributes what under which workflow context*. To further identify the most important inductive biases, Table 5 provides a finer-grained ablation of individual VCL losses and router regularizers. Removing $\mathcal{L}_{cont}$ or $\mathcal{L}_{cons}$ causes the largest degradation among VCL terms, suggesting that cross-stage functional continuity and anchor-based representation consistency are particularly important for workflow coherence. For the router, removing $\mathcal{L}_{balance}$ hurts more than removing $\mathcal{L}_{entropy}$, and removing both leads to the largest router-side drop, showing that balanced expert utilization and confident routing are complementary.

**Hyperparameter Sensitivity.** We conduct a sensitivity analysis of FedWave on Automotive (Figure 2). **VCLayer weights.** Sweeping $\lambda_{\text{pos}}$, $\lambda_{\text{cont}}$, and $\lambda_{\text{cons}}$ shows a peak near the default setting and smooth degradation as weights decrease, suggesting stable gains rather than fragile tuning. **Core hyperparameters.** We vary the SFT–VC trade-off $\gamma$, the shared VCLayer scale ($\lambda$-scale), router top-$k$, and DPO strength $\beta$; the default remains near-optimal and curves

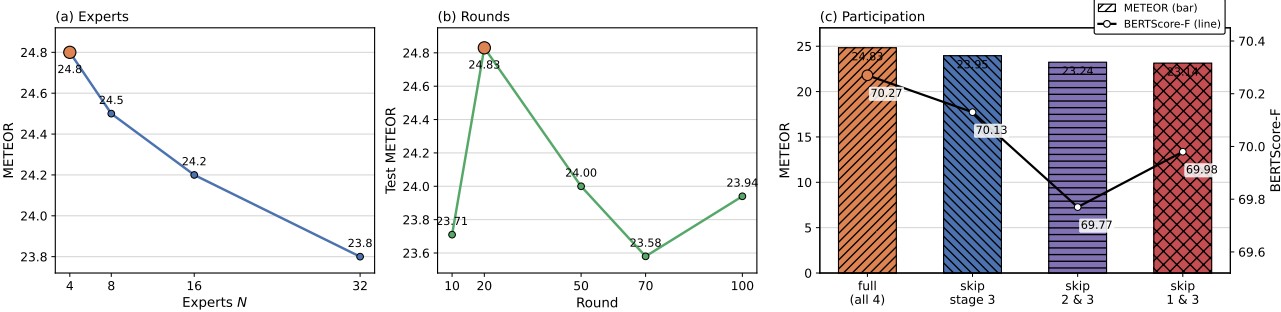

*Figure 3.* Scalability and federated protocol robustness on Automotive (Llama3-8B). (a) Scaling the number of experts $N$. (b) Longer schedules in terms of FL rounds. (c) Partial participation by skipping stages, reporting METEOR (bars) and BERTScore-F (line).

*Table 7.* Performance of the FedWave framework when integrated with different federated optimization algorithms. Experiments are conducted with the Llama3-8B backbone.

| Aggregation Algorithm | BS-F | Meteor | Rouge-L |
|---|---|---|---|
| FedWave (w/o DPO) | 66.55 | 18.40 | 17.14 |
| + FedAvgM | 70.59 | 23.43 | 21.73 |
| + FedProx | **70.86** | 23.71 | **22.20** |
| + FedAdam | 70.68 | 23.68 | 21.89 |
| + FedAdagrad | 70.71 | **23.79** | 21.97 |
| + FedYogi | 70.35 | 23.42 | 21.46 |

change gradually without sharp cliffs. **Joint grid check.** Figure 2(d) shows limited interaction among $\lambda$-scale, $\gamma$, and $\beta$, with METEOR staying within a narrow band around the default. Overall, FedWave is robust to both individual and coupled hyperparameter changes.

### 4.5. Scalability and Federated Protocol Robustness

**Robustness to deployment factors.** Figure 3 evaluates FedWave under three factors that often arise in deployments. **Scaling the number of experts.** As the expert count increases from $N=4$ to $N=32$, METEOR decreases moderately ($24.8 \rightarrow 23.8$), suggesting FedWave scales without catastrophic degradation, though larger $N$ makes routing/aggregation harder and can dilute per-expert data and updates. **Longer training schedules.** Varying FL rounds shows a clear peak at 20 rounds (24.83); extending training can reduce performance, consistent with accumulated client drift and overfitting to local objectives, so a moderate number of rounds yields a better global trade-off. **Partial participation and missing stages.** When stages are skipped, both METEOR and BS-F drop but remain relatively stable, with full participation best, indicating robustness to partial participation while performance naturally degrades when key upstream/midstream expertise is absent.

**Scaling clients within each workflow role.** To distinguish workflow-role scaling from cross-silo client scaling,

we conduct an additional experiment in which the number of workflow roles is fixed to four, while each role is further split into multiple federated clients. As shown in Table 6, FedWave remains stable when scaling from 4 clients to 8, 16, and 32 clients. Even when only 4 clients participate in each round, the METEOR score stays within a narrow range of 25.07–25.31 and does not degrade relative to the original one-client-per-role setting. This suggests that FedWave is not limited to a small 4-expert co-training setup. Instead, it can support multiple cross-silo clients under the same workflow role through role-wise aggregation.

**Compatibility with federated optimizers.** To assess modularity, we integrate FedWave with federated optimization protocols, including FedAvgM, FedProx, FedAdam, FedAdagrad, and FedYogi, while keeping all other components fixed. Table 7 shows comparable performance across optimizers, with FedProx and adaptive methods (FedAdam/FedAdagrad) competitive. The strong default suggests VCLayer and router-induced specialization reduce client drift and mitigate non-IID instability in this structured setting. Overall, FedWave is compatible with a range of federated optimizers, allowing practitioners to choose based on system constraints (e.g., stability vs. tuning complexity).

## 5. Conclusion

We introduced **FedWave**, a federated multi-agent collaboration framework for *workflow-structured* sequential tasks over decentralized, privacy-sensitive data silos. FedWave makes workflow dependencies explicit in federated training and deployment, combining a collaborative *VCLayer* for stage-aware constraints, role-specific LoRA adaptation, an *input-conditioned MoE router* for task-aware expert fusion, and a *DPO* stage that aligns end-to-end outputs using router-induced preferences. Experiments across multiple datasets, metrics, and LLM backbones show that FedWave achieves strong gains over federated baselines on the main workflow benchmarks and remains competitive with centralized multi-agent systems under realistic privacy constraints.

# Acknowledgements

This work was partially supported by grants from the National Natural Science Foundation of China (Nos. T2541001, 52472316, 52341203, and 52461160297) and the Guangdong Basic and Applied Basic Research Foundation (No. 2025A1515010251).

# Impact Statement

FedWave aims to enable privacy-preserving collaboration among organizations whose workflow data cannot be centrally pooled due to legal, compliance, or IP constraints. A positive impact is lowering the barrier for cross-silo learning in safety-critical pipelines (e.g., manufacturing and healthcare workflows) while reducing the need to transfer sensitive records. Potential risks include misuse for coordination in harmful activities, and privacy leakage through shared model updates in adversarial settings. Our approach does not provide formal privacy guarantees; it inherits known FL risks such as gradient-based inference attacks. In practice, FedWave should be deployed with complementary protections (e.g., secure aggregation, access control, audit logging, and optionally differentially private optimization), and with governance over what prompts/responses may be used for preference alignment.

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

# A. Additional Analyses on Stability, Routing, Scalability, and Practicality

This appendix provides additional analyses that complement the main experiments. We first clarify the notation used in the framework figure and examine the design choices behind the VCLayer. We then analyze routing behavior, preference construction, scalability, non-linear workflow structures, and system efficiency.

## A.1. Notation Used in Figure 1

For clarity, Table 8 summarizes the main symbols used in Figure 1. The superscript $t$ denotes the current federated round, and $t + 1$ denotes the updated parameters after local training and server aggregation.

*Table 8.* Notation used in Figure 1.

| Symbol | Meaning |
|--------|---------|
| $W_g^t$ | Shared global weights at round $t$ |
| $W_i^t$ | Local weights of expert $i$ at round $t$ |
| $W_g^{t+1}$ | Updated shared global weights after aggregation |
| $W_i^{t+1}$ | Updated local weights of expert $i$ after local training |
| $C_N$ | Top-ranked candidate response |
| $C_i$ | Lower-ranked response from expert $i$ |

## A.2. Validity of the Continuity and Consistency Proxies

*Table 9.* Comparison between the proposed continuity/consistency proxies and simpler alternatives on the Automotive workflow using the Llama3-8B backbone. We report METEOR as the primary metric.

| Method | METEOR |
|--------|--------|
| Full FedWave | **24.83** |
| w/o $\mathcal{L}_{cont}$ | 24.30 |
| $\mathcal{L}_{cont} \rightarrow$ feature alignment | 24.14 |
| w/o $\mathcal{L}_{cons}$ | 24.28 |
| $\mathcal{L}_{cons} \rightarrow$ output alignment | 24.07 |
| Both replacements | 24.09 |

Table 9 evaluates whether the proposed continuity and consistency proxies are preferable to simpler alternatives. Removing either $\mathcal{L}_{cont}$ or $\mathcal{L}_{cons}$ degrades performance, confirming that both terms are useful. Replacing $\mathcal{L}_{cont}$ with direct feature alignment or replacing $\mathcal{L}_{cons}$ with output-level alignment leads to lower METEOR scores, and replacing both terms further reduces performance. These results suggest that the proposed proxy design provides a more effective inductive bias than directly aligning intermediate features or final outputs.

## A.3. Effect of Previous-Round Snapshots

In the VCLayer, the consistency loss aligns neighboring stages using the previous-round global snapshot. This design follows the standard FL protocol, where clients perform local updates independently within each round and synchronize only after local training. Using current-round adjacent-stage states would require additional within-round cross-silo synchronization, which increases communication and weakens the parallelism of federated optimization. We therefore use the previous-round snapshot as an FL-compatible approximation to cross-stage consistency. To examine whether this delayed snapshot introduces harmful staleness, we compare it with several alternatives in Table 10. We include a variant without $\mathcal{L}_{cons}$, a warm-up strategy that disables $\mathcal{L}_{cons}$ in the first five rounds, an EMA snapshot, and a synchronized current-round snapshot that uses fresher adjacent-stage states but requires additional synchronization.

The previous-round snapshot performs very close to the synchronized current-round variant, with only a 0.05 METEOR difference, while avoiding extra within-round synchronization. It also clearly outperforms removing $\mathcal{L}_{cons}$, indicating that the consistency term remains useful even when computed from delayed adjacent-stage snapshots. The warm-up and EMA variants do not improve over the default strategy, suggesting that the one-round delay does not introduce visible convergence instability in this setting. Overall, the previous-round snapshot provides a practical trade-off between cross-stage consistency and FL-compatible communication.

*Table 10.* Effect of different snapshot strategies for the consistency loss on the Automotive workflow using the Llama3-8B backbone. We report METEOR as the primary metric.

| Snapshot strategy | METEOR |
|---|---|
| Previous-round snapshot (ours) | 24.83 |
| w/o $\mathcal{L}_{cons}$ | 24.28 |
| Warm-up (5 rounds) | 24.24 |
| EMA snapshot | 24.74 |
| Synchronized current-round snapshot | **24.88** |

## A.4. Router Preference Quality and Conditional Routing

**Router preference quality.**    To verify that the router-induced preferences used in DPO are informative rather than noisy, we measure whether routing scores align with response quality. On the E-commerce dev set (Qwen2-7B), for each prompt we sample responses from all experts, compute BERTScore-F for each response, and compare the metric ranking with the router weights. As shown in Table 11, the router's top-1 expert matches the metric-best expert for 72% of examples, and the routing weights exhibit a positive Spearman correlation with BERTScore-F, indicating that the router tends to prioritize experts that yield higher-quality outputs.

**Conditional routing behavior.**    We further analyze whether the router captures *input-conditioned specialization* rather than collapsing to a fixed expert. On the Automotive dev set, we group prompts by coarse workflow categories and report the fraction of examples where each expert is selected as top-1. Table 12 shows clear routing shifts across categories, consistent with the intended role specialization: design-related prompts prefer upstream experts, manufacturing prompts prefer midstream experts, and inspection prompts prefer downstream experts.

*Table 11.* Router preference vs. response quality on the E-commerce dev set (Qwen2-7B). Top-1 match measures how often the router's top expert is also the metric-best expert.

| Metric | Top-1 match ↑ | Spearman($\alpha$, metric) ↑ |
|---|---|---|
| BERTScore-F | 0.72 | 0.68 |

*Table 12.* Conditional routing on the Automotive dev set: fraction of examples where each expert is selected as top-1 by the router (4 experts).

| Category / Expert | E1 | E2 | E3 | E4 |
|---|---|---|---|---|
| A: vehicle design | 0.52 | 0.18 | 0.10 | 0.20 |
| B: manufacturing / production | 0.19 | 0.49 | 0.12 | 0.20 |
| C: quality inspection & testing | 0.15 | 0.18 | 0.09 | 0.58 |

## A.5. Router Behavior under Heterogeneous Role Data

FedWave trains the MoE router locally together with role-specific modules and aggregates the router parameters on the server. Since clients may hold highly role-specific data, we further examine whether the aggregated router remains informative under controlled heterogeneity. We vary the dominant share of role-specific data and report the resulting performance in Table 13.

The results show that performance degrades moderately rather than collapsing as heterogeneity increases. Even under the extreme setting with a dominant share of 0.97, FedWave still achieves 23.61 METEOR. This suggests that the aggregated router preserves useful input-conditioned specialization under severe non-IID role data.

We further inspect whether routing collapses to a single expert. Table 14 reports the minimum and maximum top-1 routing share of each expert across the controlled heterogeneity settings.

The router does not collapse to a single expert: all experts maintain non-trivial top-1 shares across the evaluated heterogeneity levels. The routing entropy also stays in a moderate range, $H(\alpha) = 1.23 \sim 1.31$, indicating that the router is neither

*Table 13.* Router robustness under controlled heterogeneity on the Automotive workflow using the Llama3-8B backbone. The dominant share indicates the fraction of data assigned to the dominant role.

| Heterogeneity level | Dominant share | METEOR |
|---|---|---|
| Low | 0.40 | 23.97 |
| Medium | 0.65 | **24.18** |
| High | 0.85 | 23.68 |
| Extreme | 0.97 | 23.61 |

*Table 14.* Top-1 routing share across experts under controlled heterogeneity. The min/max values are computed across the heterogeneity levels in Table 13.

| Expert | Top-1 share (min / max) |
|---|---|
| E1 | 0.24 / 0.29 |
| E2 | 0.21 / 0.26 |
| E3 | 0.23 / 0.27 |
| E4 | 0.20 / 0.25 |

over-confidently collapsed nor purely uniform. These observations support that global router aggregation, together with the load-balancing and entropy regularizers, can maintain stable routing behavior under heterogeneous role data.

### A.6. Robustness of Router-Induced Preference Construction

The DPO stage uses router-induced preference pairs, where the response from the top-ranked expert is treated as the preferred response and the response from a lower-ranked expert is treated as the dispreferred response. This preference signal is not assumed to be noise-free. Rather, it serves as weak collaborative supervision after federated SFT. We analyze whether more conservative pair construction strategies improve over the default design.

Let $m = \alpha_{(1)} - \alpha_{(2)}$ denote the router confidence margin between the top-1 and top-2 experts. Table 15 compares the original DPO construction with a top-1-vs-bottom-1 variant and a confidence-filtered variant that trains only on high-confidence pairs with $m \geq 0.15$.

The original DPO construction achieves the best overall METEOR. The confidence-filtered variant performs slightly better on the low-confidence subset but does not improve the overall score, suggesting that discarding lower-confidence pairs also removes useful training coverage. The top-1-vs-bottom-1 strategy is also slightly worse than the original construction, indicating that overly easy negatives are not necessarily more effective for alignment. These results suggest that router-induced preferences are noisy but useful as weak supervision. We further isolate the disagreement subset, where the router top-1 expert is not the metric-best expert. This setting directly evaluates whether router misranking causes severe error amplification during DPO.

Even on the disagreement subset, Original DPO reaches 24.21 METEOR, far above the model without DPO. This indicates that router misranking introduces noise, but does not cause catastrophic amplification in our experiments. A likely reason is that DPO is regularized by the frozen SFT reference model, so the policy is not allowed to drift arbitrarily toward noisy preferences. Overall, the results support the view that router-induced DPO provides an effective weak-supervision signal while remaining robust to moderate preference noise.

### A.7. Dataset Extension to NonLinear Workflows: Automotive-Energy

**Why Automotive-Energy is different.** The main paper evaluates FedWave on 4-stage *linear* workflows, where dependencies primarily follow a chain structure. To stress-test robustness under more realistic value-chain structures, we extend evaluation to **Automotive-Energy**, a harder workflow dataset from MSCoRe featuring *six* stages with *cross-stage coupling*. Unlike the earlier datasets where each stage maps cleanly to a single downstream stage, Automotive-Energy contains many-to-many dependencies and feedback-like interactions, where upstream decisions may be revisited after downstream signals (e.g., generation and storage constraints influencing production, or usage signals feeding back to design). This setting better reflects practical industrial workflows and provides a stronger test for workflow-aware federation.

*Table 15.* Effect of different router-induced preference pair construction strategies on the Automotive workflow using the Llama3-8B backbone. We report overall METEOR and split results by router confidence margin.

| Pair construction | Overall METEOR | $m \geq 0.15$ | $m < 0.15$ |
|---|---|---|---|
| w/o DPO | 18.40 | – | – |
| Original DPO | **24.83** | **25.17** | 23.95 |
| Top-1 vs bottom-1 | 24.63 | 24.85 | 24.05 |
| Confidence-filtered DPO | 24.77 | 24.99 | **24.19** |

*Table 16.* DPO performance on agreement and disagreement subsets. The disagreement subset contains examples where the router top-1 expert is not the metric-best expert.

| Method | Overall METEOR | top-1 = best | top-1 $\neq$ best |
|---|---|---|---|
| Original DPO | **24.83** | **24.99** | 24.21 |
| w/o DPO | 18.40 | – | – |

**Single-pass generation with workflow-structured inputs.** FedWave handles cross-stage coupling through single-pass generation rather than explicit multi-round agent interaction at inference time. Each test instance is still formulated as a question-answering example: the model receives one workflow-derived input and produces one final answer. The input is organized around the underlying workflow context, so information associated with coupled stages can appear in the same problem instance and provide cues for relevant cross-stage feedback. For example, in Automotive-Energy, a design-related question may involve battery capacity, fast-charging mode, and retired-battery recycling, while its answer should also account for electricity-price forecasts, renewable-energy penetration, storage coordination, and downstream usage feedback. Similarly, a production-related question may need to consider regional charging demand and peak-valley grid scheduling, even if these dependencies are not presented as manually enumerated constraint fields. FedWave learns to use such dependencies during federated training: the VCLayer imposes relational biases over workflow edges, role-specific LoRA modules preserve stage-specialized knowledge, and the MoE router performs input-conditioned expert fusion. Thus, single-pass generation uses workflow-structured inputs to encode relevant cross-stage feedback, and the learned workflow-aware representations further internalize these dependencies without requiring iterative communication among agents at test time.

**Results.** As shown in Table 17, FedWave consistently outperforms a range of standard FL optimizers on this non-linear workflow, demonstrating that the proposed Value Chain Layer and routing-based coordination are not restricted to simple chains and remain effective under cross-stage coupling.

*Table 17.* Performance on the non-linear Automotive-Energy workflow (Llama3-8B).

| Method | BLEU-4 | ROUGE-L | BERTScore-F |
|---|---|---|---|
| FedAdam | 0.97 | 6.05 | 53.05 |
| FedAvg | 4.19 | 9.32 | 56.80 |
| FedAvgM | 1.61 | 6.59 | 53.72 |
| FedProx | 3.16 | 8.24 | 55.69 |
| FedYogi | 0.92 | 5.88 | 52.95 |
| SCAFFOLD | 3.90 | 9.50 | 57.07 |
| **FedWave (ours)** | **20.65** | **27.62** | **76.08** |

## A.8. Expanded Expert Scaling Across Domains and Backbones

Table 18 extends the expert-scaling analysis beyond the original single-setting result. Increasing the number of experts from 8 to 16 does not cause consistent degradation across datasets or backbones. In several cases, performance even improves, suggesting that FedWave is not tied to the smallest 4-expert configuration. Together with the main scalability analysis, these results provide additional evidence that FedWave can remain stable under larger workflow expert sets.

*Table 18.* Expanded expert-scaling results across three workflow datasets and two backbones. We report METEOR for 8 and 16 experts.

| Dataset | Qwen2-7B (8) | Qwen2-7B (16) | Llama2-7B (8) | Llama2-7B (16) |
|---|---|---|---|---|
| Automotive | 36.38 | 35.79 | 25.83 | 27.19 |
| E-commerce | 44.90 | 45.49 | 21.98 | 24.14 |
| Pharmaceutical | 34.23 | 34.93 | 18.81 | 21.45 |

*Table 19.* Efficiency analysis: trainable parameters and communication overhead.

| Method | Trainable Params/client | Comm/client/round (MiB)↓ | Rel. to LoRA FedAvg |
|---|---|---|---|
| Full-model FedAvg (hyp.) | $\approx 8.0$B | $\approx 30{,}518$ | $\approx 95.4\times$ |
| FedAvg-LoRA (fair) | 83.89M | 320.0 | $1.0\times$ |
| **FedWave (ours)** | 167.83M | 640.2 | $\approx 2.0\times$ |

## A.9. Communication Overhead

**Setup.** We quantify training-time communication cost by transmitted *trainable* parameters per round (one upload plus one download per client, 16-bit). As shown in Table 19, FedWave introduces moderate overhead compared to a LoRA-only federated baseline due to the additional VCLayer and router parameters, but remains orders of magnitude smaller than full-model federated training.

## A.10. Inference-Time Efficiency vs. Centralized Multi-Agent Baselines

**Setup.** For inference-time comparisons against centralized multi-agent baselines, we follow their standard Qwen2-7B setup and report end-to-end runtime per query, counting all multi-round interactions and intermediate generations. Token counts are reported as average input/output lengths aggregated over all turns for multi-agent baselines, while FedWave uses a single forward pass.

**Result.** Table 20 shows that FedWave is substantially faster and more token-efficient than centralized multi-agent baselines that rely on long multi-turn dialogues, while maintaining competitive quality in the main results.

*Table 20.* Efficiency analysis: runtime and average input/output lengths (Qwen2-7B).

| Method | Time (s)↓ | Avg.Input | Avg.Output |
|---|---|---|---|
| Automotive (Qwen2-7B) | | | |
| PMC | 241.56 | 11,637.94 | 6,644.22 |
| MedAgents | 196.53 | 3,901.03 | 3,483.24 |
| Debate (short) | 187.16 | 3,938.35 | 4,809.27 |
| Debate (long) | 162.57 | 4,357.87 | 5,278.53 |
| CoA | 227.21 | 11,428.06 | 5,446.62 |
| **FedWave (ours)** | **38.11** | **94.16** | **1,040.31** |
| Pharmaceutical (Qwen2-7B) | | | |
| PMC | 198.49 | 3,796.07 | 2,613.73 |
| MedAgents | 169.53 | 3,504.85 | 3,504.85 |
| Debate (short) | 548.66 | 3,372.56 | 3,772.79 |
| Debate (long) | 574.95 | 3,810.87 | 4,268.74 |
| CoA | 253.18 | 5,493.94 | 3,450.45 |
| **FedWave (ours)** | **70.16** | **76.22** | **1,091.33** |

## B. Qualitative Analysis: A Case Study

To provide a more intuitive understanding of the performance differences, we present a qualitative case study on a representative task from the Automotive workflow dataset. We prompted the models with a complex, two-part request that requires both marketing creativity (Expert 1's domain) and supply chain knowledge (Expert 2's domain). The results, shown in Figure 4, highlight the distinct collaborative capabilities of each approach.

The analysis of the outputs in Figure 4 reveals the clear superiority of our proposed framework. The **FedAvg**, lacking any mechanism for structured collaboration, produces a convoluted and generic response. It struggles to differentiate between the distinct tasks of marketing and supply chain planning, mixing concepts and failing to provide the specific, actionable details required by either role. This output exemplifies the shortcomings of simple model averaging, which dilutes expert knowledge into a generalized but ultimately ineffective model.

In contrast, the **Centralized MAS** baseline, with full data access, performs well. It correctly separates the two sub-tasks and provides relevant, technically sound details for each, such as focusing on the core chassis components. This demonstrates the effectiveness of the multi-agent paradigm when privacy is not a concern.

Most importantly, **FedWave** achieves a remarkable level of performance that is highly competitive with the centralized approach, despite operating under strict privacy constraints. The output demonstrates a clear and seamless transition between the two roles. The marketing announcement is creative, while the supply chain requirements are precise and technical. Crucially, the generated details are not only plausible but also qualitatively different from the centralized model's output—for instance, specifying innovative components like 'BladeCell' batteries and 'WaveDrive' motors, and focusing on sustainable interior textiles. This suggests that `FedWave` successfully synthesizes specialized knowledge learned from distributed data sources, a feat not possible with simple averaging. This successful role-switching is a direct result of our **MoE router** dynamically identifying the relevant expert for each part of the prompt. Furthermore, the high quality and coherence of the combined output showcase the effectiveness of the **VCLayer** in maintaining a logical flow and the **DPO** stage in refining the final generation to align with high-quality, human-like collaborative solutions. This case study compellingly illustrates that FedWave successfully instills genuine collaborative intelligence in a decentralized environment.

| Input Prompt |
|---|
| *"Based on the latest market analysis indicating a strong demand for sustainable family vehicles, draft a marketing announcement for our new electric SUV, the 'EcoVoyager'. Then, outline the key component requirements for the initial production run that the supply chain team needs to source."* |

| Model | Generated Output |
|---|---|
| **Centralized MAS** (Full Data Access) | *Marketing Announcement:*

*Introducing the all-new EcoVoyager! Experience the future of family adventure with zero emissions. Designed for the conscious modern family with an exceptional range and advanced safety features. Pre-orders open this fall. #EcoVoyager #ElectricSUV #SustainableDriving*

*Supply Chain Requirements:*

*For the initial production run of 10,000 units, the key components to source are:*

*1. Battery Packs: High-capacity lithium-ion battery packs meeting our long-range specifications.*

*2. Electric Motors: High-efficiency dual-axle permanent magnet motors for the all-wheel-drive system.*

*3. Infotainment System: The latest-generation central touchscreen with our proprietary OS.*

*4. Chassis Components: Lightweight, high-strength aluminum alloy for the vehicle's unibody frame.* |
| **FedAvg** | *Marketing and Supply Chain Plan:*

*Unveil your family's next chapter with the EcoVoyager! Fusing sustainable luxury with thrilling performance, our all-electric SUV is here. Featuring a panoramic sunroof and an interior crafted from premium eucalyptus fiber, the EcoVoyager delivers an impressive range on a single charge. Drive the change. #EcoVoyager #FutureIsElectric #FamilySUV* |
| **FedWave** (Privacy-Preserving) | *Marketing Announcecment:*

*Unveil your family's next chapter with the EcoVoyager! Fusing sustainable luxury with thrilling performance, our all-electric SUV is here. Featuring a panoramic sunroof and an interior crafted from premium eco-friendly materials, the EcoVoyager delivers a 300+ mile range on a single charge. Drive the change. #EcoVoyager #FutureIsElectric #FamilySUV*

*Key Component Requirements:*

*Supply chain priority sourcing for the initial 10,000-unit production run of the EcoVoyager:*

*1. Battery System: High-density 'BladeCell' battery modules with advanced integrated thermal management.*

*2. Drivetrain: Next-generation 'WaveDrive' axial flux synchronous motors for superior power delivery (front and rear).*

*3. Central Computing Unit: The 'VisionCore AI' automotive-grade processor for infotainment and driver-assist features.*

*4. Sustainable Textiles: Sourcing of certified eucalyptus fiber textiles for all seating and interior trim..* |

*Figure 4.* Qualitative comparison of model outputs for a multi-stage task. The prompt requires the model to first act as a marketing expert and then as a supply chain analyst.

