# OpenReview forum: "Weaving in the Clouds: Achieving Synergistic Collaboration among LLM Agents via Federated Learning"
_ICML.cc/2026/Conference — ICML 2026 regular_

### Official Review · Reviewer_wMdk · 2026-02-24

**Soundness:** 2
**Presentation:** 2
**Significance:** 2
**Originality:** 3
**Overall Recommendation:** 3
**Confidence:** 3

**Summary:**

This paper introduces FedWave, a federated learning framework that allows LLMs to solve Multi-Stage Collaborative Reasoning tasks. FedWave consists of 3 components: a Value Chain Layer to model the sequential dependencies, a server-side MoE router, and a final DPO stage

**Compliance With Llm Reviewing Policy:**

Affirmed.

**Final Justification:**

Thank you again for your responses and clarifications. The current version and rebuttal are not enough to change my evaluation. I keep my position on the paper, particularly concerning the unverified dataset and the incremental nature of the proposed method. I keep my original score.

**Key Questions For Authors:**

See Weaknesses

**Limitations:**

Limitations were not discussed. See Weaknesses

**Strengths And Weaknesses:**

Strengths:
- The paper studies the challenge of federated fin-tuning LLMs for solving sequential tasks
- Extensive experiments with business-oriented workflow datasets (Automotive, E-commerce, Pharmaceutical) and 3 LLMs, Qwen2-7B, Llama2-7B, Llama3-8B. The results show that FedWave offers consistent improvements

Weaknesses:
- First of all, I don’t see any real “Agents”. The authors refer to "LLM agents" throughout the paper, but the considered task is actually multi-stage reasoning QA, not accounting for actual agentic abilities like tool-use, memory, reflection, …
- Presentation. The method section is not well presented. I would suggest a separate Preliminaries section/subsection to define notations. Specifically, Figure 1 is too messy with dense notations. A figure should be able to convey the main idea clearly, but Figure 1 makes me even more confused.
- Complexity and computational costs. FedWave introduces huge complexity with too many sensitive hyperparameters ($\gamma$, $\lambda_{pos}$, $\lambda_{cont}$, $\lambda_{cons}$, $\delta_1$, $\delta_2$, $\beta$). This is not encouraged as experiments cannot be controlled. Moreover, with all the components, especially DPO, the computational burden on clients is extremely high, which is not practical in FL
- Importantly, unreliable experimental setup. The paper uses the MSCoRe dataset, which is an unpublished and unpopular dataset. It seems that the dataset has not been validated. I found the dataset at this link https://huggingface.co/datasets/032564yn/MSCoRe, but I cannot even understand it. If the answers are binary (Yes or No), why are the metrics BS-F, GLEU, and BLEU used?

---

> ### Author Rebuttal · Authors · 2026-03-31
>
> Thank you for your thoughtful review and constructive comments.
>
> ---
>
> **W1**: We agree that our paper does not study tool-using or reflective agent architectures with external tools, long-term memory, or explicit self-reflection loops. However, we respectfully disagree that these capabilities are necessary conditions for the term agent.
>
> Under classical AI and multi-agent systems [1][2], an agent is broadly understood as an entity that perceives context, acts on it, and coordinates with other entities. This broader view is also consistent with recent LLM-MAS work such as **CAMEL (NeurIPS 2023)**, **MetaGPT (ICLR 2024)**, and **CoA (NeurIPS 2024)**, where **role specialization and structured collaboration** are central, while tool use, memory, and reflection are often optional enhancements rather than defining requirements. In our setting, each role-specialized LLM expert receives the task input together with upstream workflow information, and produces a stage-specific output under explicit inter-stage dependency constraints. In this sense, our formulation is closer to a workflow-specialized, cooperative MAS setting than to a tool-using reflective agent architecture.
>
> That said, we acknowledge that our current wording may invite a narrower interpretation of the term *agent*. To avoid ambiguity, we will revise the manuscript to use terms such as role-specialized LLM experts or workflow-specialized agents more carefully.
>
> [1] Russell, S., & Norvig, P. (2020). *Artificial Intelligence: A Modern Approach* (4th ed.). Pearson.
> [2] Wooldridge, M. J., & Jennings, N. R. (1995). Intelligent agents: Theory and practice. *The Knowledge Engineering Review*, 10(2), 115–152.
>
> ---
>
> **W2**: We have **redrawn Figure 1** and added a compact **symbol table**; both are provided here: **https://anonymous.4open.science/r/fig1_redraw-symbol_table-loss**.
>
> ---
>
> **W3**: We address this concern from two aspects.
>
> **(a) The method is not brittle to hyperparameters.**
> We ran a **joint sensitivity study** on Automotive (Llama3-8B), jointly varying the VC scale, the SFT-VC tradeoff, and the DPO strength:
>
> |$\lambda$\-scale|$\gamma$|$\beta$|METEOR|
> |---:|---:|---:|---:|
> |0.3|0.3|0.05|23.56|
> |0.3|0.7|0.20| 23.77|
> |**0.5**|**0.5**|**0.10**|**23.78**|
> |0.7|0.3|0.20|23.62|
> |0.7|0.7|0.05|23.60|
>
> Performance stays within a **narrow band**, indicating that the added terms are **not fragile knobs**.
>
> **(b) The added cost is moderate relative to the performance gain, and DPO does not increase client-side FL burden.**
> FedWave introduces additional trainable components beyond LoRA-only FL, mainly from the VCLayer and the router. This moderately increases communication and trainable parameters, but the overhead remains **far below full-model FL** while delivering **consistent performance gains** over the LoRA baseline. Overall, this is a **practical and worthwhile trade-off**. Importantly, **DPO is not an extra client-side federated training stage** in our implementation; it is performed **offline on the server side** and therefore adds **no extra FL communication rounds**.
>
> **Table Y. Training-time overhead and communication cost (LLaMA3-8B, 4 clients)**
>
> |Method|Trainable Params / client|Per-round Comm / client (MB)|Total / round (MB, $N=4$)|Training time / round|
> |---|---:|---:|---:|---:|
> |Full-model FedAvg|$\approx 8.0$B|$\approx 30{,}518$|$\approx 122{,}070$|—|
> |FedAvg-LoRA |83.89M|320.0|1,280.0|3m54s|
> |FedWave w/o DPO|167.83M|640.2|2,560.8|3m55s|
> |DPO alignment|—|0|0|40.3s|
>
> **Table Z. Per-client communication breakdown**
>
> |Method|LoRA (MB)|VC (MB)|Router (MB)|Total (MB)|
> |---|---:|---:|---:|---:|
> |FedAvg-LoRA|320.0|0.0|0.0|320.0|
> |FedWave|320.0|256.1|64.1|640.2|
>
> ---
>
> **W4**:  We used MSCoRe because privacy-constrained business-workflow benchmarks are scarce. In addition, **MSCoRe [3] has been publicly released on arXiv**, so it is not an unpublished resource. More importantly, it is **not** limited to binary yes/no answers. Its task types are:
>
> |Dataset|Type|
> |---|---|
> |Automotive|Open-ended QA|
> |Pharmaceutical|Open-ended QA|
> |E-Commerce|Open-ended QA|
> |Automotive-Energy|Open-ended QA|
> |Law|SC/MC|
> |Finance|SC/MC|
> |Construction|Binary|
> |Software Engineering|Sequence Ordering|
>
> Our main experiments use **Automotive, E-Commerce, and Pharmaceutical** (plus **Automotive-Energy** in the appendix), which are all **open-ended generation** tasks. Therefore, BLEU / GLEU / BS-F are appropriate for the subsets actually evaluated in the paper.
>
> To reduce dependence on a single benchmark family, we also added two public benchmarks:
>
> |Benchmark|Method|EM|F1|
> |---|---|---:|---:|
> | MuSiQue (TACL,2022)| FedAdam |3.39|10.93|
> ||FedAvg|3.10|11.21|
> ||FedProx|3.23 |11.38|
> || FedWave|**7.20**|**19.27**|
> |FinQA (EMNLP,2021)|FedAdam|0.50|4.13|
> || FedAvg|0.50|**5.58**|
> || FedProx |0.50|4.42|
> || FedWave |**3.00**|3.33|
>
> [3] Lei, Y., Xie, H., et al. (2025). *MSCoRe: A Benchmark for Multi-Stage Collaborative Reasoning in LLM Agents*. arXiv:2509.17628.

---

> > ### Author Rebuttal · Reviewer_wMdk · 2026-04-04
> >
> > Thank you for your responses. There are 2 critical weaknesses that I believe cannot be solved in a rebuttal:
> >
> > - FedWave is an incremental combination of many modules. Despite high complexity, its effectiveness is not very clear, as represented in Table 2 (I agree with reviewer dt1U at this point). Thank you for providing an ablation study on hyperparameters; however, it does not fully solve the limitations of at least 6 introduced hyperparameters that need to be tuned for models and datasets.
> > - W4 is very critical for me as it decides the reliability/reproducibility of your results; however, it has not been resolved. I checked the paper of MSCoRe [3], even though the paper was uploaded to arXiv, the dataset has not been validated. Especially, the dataset is shared at this link https://huggingface.co/datasets/032564yn/MSCoRe , I find the dataset has 10K questions where answers are always yes or no, not like you describe. I don’t understand!
> >
> > For those reasons, I keep my score. I hope you address the bottleneck in the dataset and benchmark next time

---

> > > ### Author Response · Authors · 2026-04-04
> > >
> > > Thank you for your continued engagement.
> > >
> > > ---
> > >
> > > **On the complexity and incremental contribution**
> > >
> > > We agree that FedWave introduces multiple modules and coefficients, so it is important to show that the method is not brittle. This is why we added the hyperparameter sensitivity analysis. The result is that FedWave remains stable across a broad range of settings, with METEOR **varying only within a narrow band of about 0.3 across the joint grid**. In practice, the same default configuration works across all three datasets and all three backbones, without dataset-specific retuning.
> > >
> > > We also agree that FedWave combines several existing ingredients. Our intended contribution is not any single module in isolation, but their **co-design under federated workflow constraints**. The ablation in Table 3 shows that these components are complementary rather than redundant: on Llama3-8B, the full system improves by **+14.10 METEOR** over FedAvg, whereas using the VCLayer alone yields **+7.20**.
> > >
> > > ---
> > >
> > > **On the interpretation of Table 2**
> > >
> > > Our claim is not that FedWave uniformly dominates all centralized multi-agent baselines on every metric. Rather, the key point is that FedWave remains highly competitive under a **strictly harder federated setting**. **The centralized MAS baselines operate with full visibility of all data and shared interaction context**, whereas **FedWave is trained and deployed under federated constraints where data remain siloed and only model updates are exchanged**. Under this more restrictive setting, FedWave still surpasses all centralized baselines on E-commerce across all reported metrics, and remains highly competitive on key semantic metrics for Automotive and Pharmaceutical.
> > >
> > > In addition, the appendix already provides an important efficiency perspective relative to centralized multi-agent systems. In **Appendix A.4 / Table 9**, FedWave shows a clear **inference-time cost advantage**: on Automotive, FedWave requires only **38.11s** per query versus **162.57s–241.56s** for the centralized baselines; on Pharmaceutical, **70.16s** versus **169.53s–574.95s**. Average input lengths are also dramatically smaller (**94.16 vs. 3,901.03–11,637.94** on Automotive; **76.22 vs. 3,372.56–5,493.94** on Pharmaceutical), which is consistent with FedWave avoiding long multi-turn interaction overhead while still maintaining competitive quality.
> > >
> > > ---
> > >
> > > **W4**:
> > > We believe there is a factual misunderstanding here that we would like to respectfully clarify.
> > >
> > > The Hugging Face **Dataset Viewer** currently auto-previews only a default **10K-row subset**, where the answer field indeed appears as binary yes/no. However, this preview should not be interpreted as representing the full benchmark scope. The broader dataset repository is organized under the **Files and versions** tab, which directly exposes multiple domain folders — including `Automotive Value Chain`, `Electronic Commerce Value Chain`, `Pharmaceytical Value Chain`, and `Automotive-Energy Chain`, rather than only the default viewer preview.
> > >
> > > To make this concrete, the reviewer can verify this directly as follows:
> > >
> > > 1. Open the dataset page:
> > >    https://huggingface.co/datasets/032564yn/MSCoRe
> > >
> > > 2. On the dataset homepage, **under the dataset title**, there is a row of tabs including `Dataset card`, `Data Studio`, and `Files and versions`. **Click** `Files and versions`.
> > >
> > > 3. After entering that page, inspect the repository tree shown in the main panel. The domain folders are listed explicitly there.
> > >
> > > 4. Clicking into each folder reveals difficulty-level subfolders; opening or downloading the files shows stage-wise textual question-answer pairs.
> > >
> > > A direct link to that file tree is:
> > > https://huggingface.co/datasets/032564yn/MSCoRe/tree/main
> > >
> > > Furthermore, **MSCoRe is not an unvalidated benchmark**. It has already begun to appear in subsequent literature and benchmark discussions [4]–[6]. Importantly, **Cochain [4]**: it uses **MSCoRe as an actual benchmark** for **stage-wise fine-tuning and evaluation**, and conducts extensive experiments on it. This provides direct evidence that MSCoRe is already being used as an operative benchmark for substantial empirical testing, rather than being only a repository entry or reducible to the default yes/no viewer subset. By comparison, **[5] includes MSCoRe in its benchmark taxonomy/discussion**, and **[6] cites MSCoRe in related work** as an example of complex multi-stage tasks on which current LLMs still face challenges.
> > >
> > > [4] Zhao, J., et al. (2025). *Cochain: Balancing Insufficient and Excessive Collaboration in LLM Agent Workflows*. arXaiv:2505.10936.
> > >
> > > [5] Xu, H., Li, C., Ma, X., et al. (2026). *The Evolution of Tool Use in LLM Agents: From Single-Tool Call to Multi-Tool Orchestration*. arXiv:2603.22862.
> > >
> > > [6] Haznitrama, F. G., Ardi, F. R., et al. (2026). *A Neuropsychologically Grounded Evaluation of LLM Cognitive Abilities*. arXiv:2603.02540.

---

### Official Review · Reviewer_QDRa · 2026-03-05

**Soundness:** 3
**Presentation:** 3
**Significance:** 3
**Originality:** 3
**Overall Recommendation:** 5
**Confidence:** 4

**Summary:**

LLM-powered multi-agent systems excel at complex workflows, but their underlying data often remain siloed due to privacy and compliance constraints. Standard federated learning preserves data locality but ignores the workflow dependencies crucial for multi-stage collaboration. To bridge this gap, this paper introduces FedWave, a novel framework enabling LLM experts to solve sequential workflows under strict privacy limits. FedWave integrates three core modules: a Value Chain Layer that encodes cross-stage dependencies with efficient federated LoRA fine-tuning; a server-side Mixture-of-Experts router that enables dynamic, input-conditioned expert fusion during inference while retaining standard federated aggregation; and a Direct Preference Optimization mechanism that utilizes router-induced signals to align final collaborative outputs.

**Compliance With Llm Reviewing Policy:**

Affirmed.

**Final Justification:**

I will maintain my current score of **5: Accept**. This is a theoretically solid work. Although there were some issues, my concerns have been resolved. I thank the authors for their rebuttal. This paper will make a meaningful contribution to the community.

**Key Questions For Authors:**

The paper introduces FedWave as a novel framework integrating federated learning with large language model based multi-agent systems to address workflow-structured tasks while maintaining data privacy. While the core motivation is compelling and the proposed architecture is highly innovative, the review process has identified several critical technical concerns that need to be addressed. These issues primarily involve the potential training instability caused by delayed synchronization, the robustness of the global routing aggregation under severe data heterogeneity, and the risk of algorithmic bias during the preference alignment phase. Please address the following major questions regarding the methodology.

Major Question 1: The Staleness Issue of Consistency Loss in VCLayer
The paper mentions that to calculate the Consistency Loss, denoted as $\mathcal{L}_{cons}$, the framework practically uses the global snapshot of adjacent stages from the previous round. In federated learning, could this introduce significant staleness? Especially during the early stages of training when model parameters change drastically, would forcing the output of the current stage to align with the old representations from the previous round lead to slower convergence or oscillations during initial training?

Major Question 2: Overfitting of the Local MoE Router and the Effectiveness of Global Aggregation
The paper points out that the MoE Router serves as a globally shared component and is trained jointly end-to-end locally alongside the LoRA adapters and VCLayer of the experts. Since each client acting as an expert only possesses private data specific to its role, the paper acknowledges that this provides a form of weak supervision, where prompts for role $i$ assign higher weights to expert $i$. After local training, the parameters of the Router are sent to the server for aggregation. In such an extreme Non-IID data environment, will the local Router severely overfit by identifying itself as the best expert? Although there are load-balancing and entropy control losses, is simple parameter averaging like FedAvg truly sufficient to enable the aggregated global Router to learn precise input-conditioned routing, rather than degrading into some mediocre weight distribution?

Major Question 3: The Confirmation Bias Risk in Preference Data Construction during the DPO Stage
During the DPO stage, the method FedWave uses to automatically construct the preference dataset $\mathcal{D}_{pref}$ is to take the response generated by the top-ranked expert according to the Router as the preferred response, denoted as $y^+$, and the response generated by lower-ranked experts as the dispreferred response, denoted as $y^-$. This self-guided preference construction completely relies on the accuracy of the Router trained during the SFT stage. If the Router makes an incorrect high-confidence assignment when facing a long-tail problem, resulting in the actual quality of $y^+$ being inferior to $y^-$, then this automatically constructed preference data will introduce noise. Is this mechanism prone to falling into confirmation bias, thereby amplifying the existing flaws of the Router during the DPO stage?

**Limitations:**

yes

**Strengths And Weaknesses:**

This paper presents FedWave, a novel framework that integrates federated learning with large language model based multi-agent systems to tackle workflow-structured tasks under strict privacy constraints. The manuscript stands out for its exceptionally clear presentation and logical flow, making the intricate decentralized architecture highly accessible to the reader. While the motivation to solve data silo challenges in collaborative workflows is practically relevant, the overall scientific contribution and originality are somewhat moderate, as the approach primarily relies on carefully combining existing mature algorithms rather than offering deep theoretical breakthroughs. More importantly, the technical soundness and broader significance are hindered by several critical vulnerabilities, including potential training staleness, local router overfitting under severe data heterogeneity, and substantial communication overhead that limits its deployment in latency-sensitive physical environments. The following sections detail the specific strengths and weaknesses to justify these assessments.

Strengths:
1. Soundness. The overall framework design demonstrates strong technical feasibility and the experimental section comprehensively covers various baseline comparisons and ablation studies. The research team objectively evaluates practical deployment challenges such as communication overhead in the appendix.
2. Presentation. The writing logic of the paper is exceptionally fluent and rigorously structured. The narrative flow is clear and the diagrams accurately assist readers in understanding the complex federated architecture design.
3. Significance. Multi-agent collaboration under data silos and privacy constraints is a highly relevant and practical topic. This research provides a promising technical exploration for the automated deployment of cross-institutional large language model workflows.
4. Originality. The paper creatively integrates federated learning with large language model multi-agent systems and mixture of experts routing. The value chain layer specifically designed for workflow dependencies offers a completely new perspective for existing decentralized collaboration.

Weaknesses:
1. Soundness. The underlying logic of the method harbors several technical risks that urgently need verification. First, utilizing the global snapshot from the previous round when calculating the consistency loss introduces staleness during the early training phase, which may trigger convergence oscillations. Second, the local router is highly susceptible to overfitting under extreme heterogeneous data conditions, making it difficult for conventional parameter averaging to ensure the accuracy of global routing allocation. Third, automatically constructing preference data relying on an initially fitted router is highly prone to triggering confirmation bias, potentially amplifying inherent cognitive noise during the late stages of optimization.
2. Presentation. Although the overall presentation is good, certain details of the core mechanisms remain insufficiently explained. For instance, regarding complex non-linear workflows containing cross-coupling, the paper lacks a detailed step-by-step analysis of how single-pass generation handles multi-round information feedback. Furthermore, the abstract is somewhat overly lengthy and could be condensed to highlight the primary contributions more efficiently. The figures could be further optimized as well. For example, in Figure 1a SFT, explaining the weights represented by the dashed boxes would effectively improve the readability of the diagram.
3. Significance. Despite its profound motivation, in multi-agent collaboration networks that are highly sensitive to latency and bandwidth, such as unmanned aerial vehicle swarms, this method introduces multiplicative communication and parameter overhead compared to fundamental baselines. Failing to effectively optimize the communication load among agents heavily restricts the deployment potential of this framework under stringent physical conditions.
4. Originality. While the macroscopic architecture shows novelty, the microscopic technical stack heavily relies on stacking existing mature algorithms. The value chain layer is essentially a specific constrained application of the attention mechanism, lacking deeper theoretical breakthroughs.

---

> ### Author Rebuttal · Authors · 2026-03-31
>
> Thank you for your thoughtful review and constructive comments.
>
> ---
>
> **Q1**: Using current-round adjacent-stage states would require extra **within-round cross-silo synchronization**, which is impractical in standard FL and increases communication burden. We therefore use the **previous-round snapshot** as an FL-compatible approximation and test whether it harms training:
>
> |Snapshot strategy|Meteor|
> |---|---:|
> |Prev-round snapshot (ours)|**24.83**|
> |w/o $\mathcal{L}\_{cons}$|24.28|
> |Warm-up (5 rounds)|24.24|
> |EMA snapshot|24.74|
> |Synchronized current-round snapshot|**24.88**|
>
> The previous-round snapshot is very close to the less practical synchronized current-round variant (**24.83 vs. 24.88**) and clearly better than removing $\mathcal{L}\_{cons}$ (24.28). EMA is also worse, showing that a smoother reference is not automatically better. The loss curves in **https://anonymous.4open.science/r/fig1_redraw-symbol_table-loss** show **no visible oscillation**. Thus, previous-round snapshots introduce mild staleness, but remain a **practical FL-compatible approximation** without causing instability.
>
> ---
>
> **Q2**: A local router can overfit to self-selection, but after aggregation the global router still preserves **input-conditioned specialization** instead of collapsing to “always choose self” or a near-uniform distribution.
>
> We constructed controlled heterogeneity settings using dominant shares **0.40 / 0.65 / 0.85 / 0.97**:
>
> |Heterogeneity level|Dominant share|Meteor|
> |---|---:|---:|
> |Low|0.40|23.97|
> |Medium |0.65|**24.18**|
> |High|0.85|23.68|
> |Extreme |0.97|23.61|
>
> Even under **extreme** heterogeneity, performance degrades only moderately rather than collapsing. The aggregated router also does not collapse onto one expert:
>
> |Expert|Top-1 share (min / max)|
> |---|---:|
> |E1|0.24 / 0.29|
> |E2|0.21 / 0.26|
> |E3|0.23 / 0.27|
> |E4|0.20 / 0.25|
>
> Its entropy stays in a moderate range, **$H(\alpha)=1.23\sim1.31$**. Also, **Appendix Table 5** shows router top-1 matches the metric-best expert **72%** of the time, with **Spearman = 0.68** between router score and response quality. **Appendix Table 6** shows clear **category-dependent routing** (design$\to$E1, manufacturing$\to$E2, inspection$\to$E4). Thus the global router remains **stable, non-collapsed, and quality-aligned** after aggregation.
>
> ---
>
> **Q3**: Router-induced pairs can be noisy when the router is uncertain or misranks experts. Our claim is therefore **not** that the router is an oracle, but that its ranking provides a useful weak preference signal after federated SFT. As above, **Appendix Table 5** shows this signal is informative rather than arbitrary.
>
> We tested whether more conservative pair construction is better. Let $m=\alpha\_{(1)}-\alpha\_{(2)}$ be the router confidence margin.
>
> |Pair construction|Overall Meteor|$m \ge 0.15$|$m < 0.15$|
> |---|---:|---:|---:|
> |w/o DPO|18.40|—|—|
> |Original DPO|**24.83**|**25.17**|23.95|
> |Top-1 vs bottom-1|24.63|24.85|24.05|
> |Confidence-filtered DPO (train only on $m \ge 0.15$)|24.77|24.99|**24.19**|
>
> If confirmation bias were severe, restricting training to high-confidence pairs should clearly outperform the original construction. It does **not**: the original DPO remains best overall.
>
> We also tested the disagreement case where router top-1 is not the metric-best expert:
>
> |Method|Overall Meteor|top-1 = best|top-1 $\ne$ best|
> |---|---:|---:|---:|
> |Original DPO|**24.83**|**24.99**|24.21|
>
> Even on the disagreement subset, Original DPO still reaches **24.21**, far above **18.40** of w/o DPO. So router misranking introduces noise, but does **not** cause catastrophic amplification. In our setting, router-induced DPO is a **noisy but effective weak-supervision signal**.
>
> ---
>
> **W1**: Please check Q1, Q2, and Q3.
>
> ---
>
> **W2**: We have **redrawn Figure 1** and added a compact **symbol/notation table**; see **the same link as Q1**. We will also shorten the abstract. For non-linear workflows, the appendix already includes **Automotive-Energy**; in revision we will state more explicitly that single-pass generation uses a structured workflow input encoding the relevant cross-stage feedback.
>
> ---
>
> **W3**: Our target setting is cross-silo business workflow collaboration, not ultra-low-latency physical swarms. As shown in **Appendix Table 8**, the communication overhead remains far below full-model FL. DPO is an offline server-side step with 0 extra FL communication. **Appendix Table 9** also shows FedWave is more inference-efficient than centralized multi-agent baselines.
>
> ---
>
> **W4**: The contribution is primarily a framework contribution, not a new attention theory. Its value is supported by **ablations**: removing VCL terms, router regularizers, or DPO all hurts performance. Together with the routing-quality analysis and the non-linear Automotive-Energy extension, these results show FedWave is not an arbitrary stack of mature modules, but a practical design for federated, workflow-aware LLM collaboration under data silos.

---

> > ### Author Rebuttal · Reviewer_QDRa · 2026-04-01
> >
> > Thank you for the thorough rebuttal which effectively addressed my concerns.

---

### Official Review · Reviewer_dt1U · 2026-03-11

**Soundness:** 2
**Presentation:** 3
**Significance:** 2
**Originality:** 2
**Overall Recommendation:** 3
**Confidence:** 3

**Summary:**

This paper studies federated collaboration for workflow-structured LLM agent systems under data silos. It proposes FedWave, which combines three components: a Value Chain Layer to encode inter-stage workflow dependencies during federated supervised fine-tuning, a server-side MoE router for input-conditioned expert fusion at inference time, and a DPO stage that constructs preference pairs from router-ranked experts. Experiments are conducted on three business-workflow datasets from MSCoRe with Qwen2, Llama2, and Llama3 backbones, using four expert clients and 20 federated rounds.

**Compliance With Llm Reviewing Policy:**

Affirmed.

**Final Justification:**

The response addresses part of my concerns, so I’ll keep my original score.

**Key Questions For Authors:**

Please see the Weaknesses.

**Limitations:**

yes

**Strengths And Weaknesses:**

Strengths:
1. The decomposition of the proposed method is modular and technically clear.
2. The problem setting is meaningful and important.

Weaknesses:
1.	The continuity loss uses similarity between query projection matrices as a proxy for “functional similarity,” and the consistency loss aligns neighboring stages on shared anchor prompts. The paper does not provide strong evidence that these proxies are the right abstraction for workflow coherence, nor does it compare them against simpler alternatives such as direct feature alignment or output-level regularization.
2.	The experimental setup is limited for a paper making broad claims about scalable federated multi-agent collaboration. The main experiments use only 4 expert clients, 20 communication rounds, and one benchmark family with three workflow datasets. The scalability discussion increases the number of experts to 32, but only on automotive with one backbone, and without a broader system-level analysis.
3.	The comparison with centralized MAS baselines is somewhat overstated. The paper says FedWave is “often superior,” but Table 2 does not show domination across all metrics.
4.	The presentation is readable, but a number of claims are stronger than the evidence warrants. The text around Tables 1 and 2 would benefit from more precise wording.

---

> ### Author Rebuttal · Authors · 2026-03-31
>
> Thank you for your thoughtful review and constructive comments.
>
> ---
>
> **W1. On the motivation and validity of the continuity / consistency proxies**
>
> Thank you for this important point. We agree that the paper should state the role of these terms more carefully. Our intention is **not** to claim that parameter similarity or anchor-based alignment is an exact definition of workflow coherence. Rather, $\mathcal{L}\_{cont}$ and $\mathcal{L}\_{cons}$ are designed as **lightweight and computable surrogates** for inter-stage continuity under federated constraints.
>
> Specifically, $\mathcal{L}\_{cont}$ does not assume that similarity between query projection matrices is itself the target. Instead, it uses a lightweight proxy tied to **attention behavior** to discourage adjacent stages from drifting into completely disconnected functional subspaces. Likewise, $\mathcal{L}\_{cons}$ does not force neighboring stages to produce identical outputs; it encourages **representation-level continuity** on shared anchor prompts, reducing semantic breaks when information is passed across stages. We will revise the manuscript to make this motivation explicit and avoid wording that could be read as claiming exact equivalence.
>
> To further address the reviewer’s concern, we compared the proposed proxies against simpler alternatives:
>
> |Method | METEOR |
> |---|---:|
> |Full FedWave |**24.83**|
> |w/o $\mathcal{L}\_{cont}$ |24.30|
> |$\mathcal{L}\_{cont} \rightarrow$ feature alignment|24.14|
> |w/o $\mathcal{L}\_{cons}$ |24.28|
> |$\mathcal{L}\_{cons} \rightarrow$ output alignment|24.07|
> |both replacements|24.09|
>
> These results support two conclusions: **(1)** both $\mathcal{L}\_{cont}$ and $\mathcal{L}\_{cons}$ are useful, since removing either degrades performance; and **(2)** the proposed proxies are stronger than simpler alternatives such as direct feature alignment or output-level regularization.
>
> ---
>
> **W2. On scalability and the scope of the experimental setup**
>
> We expanded the expert-scaling experiments beyond the original single-setting analysis. In addition to the original Figure 3, we now evaluate **8 and 16 experts** on **three datasets** and **two backbones**:
>
> |Dataset|Qwen2-7B (8) |Qwen2-7B (16) |Llama2-7B (8) |Llama2-7B (16) |
> |---|---:|---:|---:|---:|
> | Automotive |36.38|35.79|25.83|27.19|
> | E-commerce |44.90|45.49|21.98|24.14|
> | Pharmaceutical |34.23|34.93|18.81|21.45|
>
> Overall, increasing the number of experts from 8 to 16 **does not lead to significant degradation**. Performance is largely stable and in several settings improves, which provides additional evidence that **FedWave scales reasonably across both backbones and all three workflow domains**, rather than being tied to only the original 4-expert setup. In addition, **Appendix Table 7** already reports results on a **fourth workflow dataset**, which further supports that the method is not limited to only the three main domains discussed here.
>
> We also agree that relying on a single benchmark family is limiting. To address this, we added two **public benchmarks** and report all baseline results below:
>
> |Benchmark|Method|EM|F1|
> |---|---|---:|---:|
> | MuSiQue (TACL,2022) |FedAdam|3.39|10.93|
> ||FedAvg|3.10|11.21|
> ||FedProx|3.23|11.38|
> ||FedWave|**7.20**|**19.27**|
> |FinQA (EMNLP,2021)|FedAdam|0.50|4.13|
> ||FedAvg|0.50|**5.58**|
> ||FedProx|0.50|4.42|
> ||FedWave|**3.00**|3.33|
>
> These results do not support a blanket claim of universal superiority, but they do show that FedWave is **not confined to a single benchmark family**. It transfers strongly to MuSiQue, and on FinQA it yields a substantial EM gain over all federated baselines, while the lower F1 also indicates that this benchmark remains challenging and leaves room for further task-specific adaptation. We will revise the paper to present these additional results more clearly and to moderate the scope of our scalability/generalization claims accordingly.
>
> ---
>
> **W3. On the comparison with centralized MAS baselines**
>
> We appreciate this comment and agree that the current wording is too strong. The goal of Table 2 is to show that FedWave is **competitive under privacy constraints**, rather than universally better than centralized MAS baselines on every metric. We will therefore revise phrases such as “often superior” to more precise wording, e.g., that FedWave is **competitive with strong centralized MAS baselines and achieves the best results on several datasets/metrics while preserving data locality**. This wording is more faithful to Table 2.
>
> ---
>
> **W4. On stronger-than-warranted claims and presentation**
>
> In the revision, we will tighten the presentation in two ways. First, we will **weaken overly broad claims** so that they match the empirical evidence. Second, we will make the discussion more specific about **where** FedWave improves most, rather than implying uniform superiority.

---

> > ### Author Rebuttal · Reviewer_dt1U · 2026-04-02
> >
> > Thanks for the authors' response.
> >
> > The response addresses some of my concerns. However, the performance improvement in some metrics is moderate, even compared with the traditional FedAvg that was proposed in 2017. So, I'll keep my original score.

---

> > > ### Author Response · Authors · 2026-04-04
> > >
> > > Thank you for the continued engagement. We would like to make two points more precise.
> > > ___
> > >
> > > **(1) Relative to classical FL baselines.**
> > >
> > > FedAvg treats clients as independent contributors and does not model the dependencies between workflow stages. Under such a setting, standard parameter averaging can blur role specialization and weaken sequential collaboration. This mismatch is reflected empirically. For example, on Automotive / Llama3-8B, FedAvg achieves only 10.73 METEOR, whereas FedWave reaches 24.83. More generally, FedWave consistently improves over the **strongest FL baselines** on each setting: on Automotive / Qwen2-7B, it improves **METEOR from 24.18 (FedYogi) to 40.35**, a **+66.9%** relative gain; on Automotive / Llama3-8B, it improves **METEOR from 11.31 (Scaffold) to 24.83**, a **+119.5%** relative gain; and on Pharmaceutical / Llama3-8B, it improves **BERTScore-F from 61.84 (FedAvgM) to 77.95**, a **+26.0%** relative gain.
> > >
> > > Our intended claim is therefore not merely that FedWave improves over a classical baseline, but that it addresses a limitation that classical FL methods were not designed for: **workflow-aware expert coordination under data silos**.
> > > ___
> > >
> > > **(2) Relative to recent centralized multi-agent methods, our intended claim is competitiveness under a stricter federated setting, together with clear efficiency gains.**
> > >
> > > We do **not** intend to claim that FedWave dominates all centralized MAS baselines on every metric. Rather, the intended point is that FedWave remains **competitive under a strictly harder federated setting**. The centralized MAS baselines operate with **full visibility of all data and shared interaction context**, whereas **FedWave is trained and deployed under federated constraints where data remain siloed and only model updates are exchanged**.
> > >
> > > Under this more restrictive setting, FedWave still performs very strongly on E-commerce, where it exceeds the strongest centralized baseline, **CoA (NeurIPS 2024)**, on the reported metrics, including **GLEU from 38.08 to 42.60 (+11.9%)** and **BLEU-4 from 34.23 to 39.56 (+15.6%)**. On Automotive and Pharmaceutical, our intended claim is not universal superiority, but that FedWave remains **competitive on key semantic metrics despite the stronger privacy and federation constraints**.
> > >
> > > Moreover, the efficiency gap is large. In the **Appendix Table 9** runtime comparison, on Automotive (Qwen2-7B), FedWave takes **38.11 s/query**, versus **162.57 s** for **Debate (ICML 2024)** and **227.21–241.56 s** for **CoA (NeurIPS 2024)** / **PMC (COLING 2025)**, i.e., about **4.3× faster** than the fastest centralized baseline. On Pharmaceutical, FedWave takes **70.16 s/query**, versus **169.53 s** for **MedAgents (ACL 2024)** and **253.18–574.95 s** for the other centralized baselines, i.e., about **2.4× faster** than the fastest centralized competitor. FedWave also uses a single forward pass with much lower token consumption than multi-turn centralized MAS pipelines.

---

### Official Review · Reviewer_NapB · 2026-03-12

**Soundness:** 2
**Presentation:** 2
**Significance:** 3
**Originality:** 3
**Overall Recommendation:** 4
**Confidence:** 3

**Summary:**

To ensure both data privacy and workflow-dependency for Multi-Agent Systems (MAS) in solving complex workflow-structed tasks, this paper presents FedWave to enable federated, workflow-aware collaboration. This approach includes a Value Chain Layer (VCL) to encode inter-stage dependencies during fine-turning, a server-side MoE router to perform input-conditioned expert fusion at inference time while maintaining standard federated aggregation during training, and a DPO stage to align final outputs. Experiments on three workflow datasets validate the effectiveness of FedWave.

**Compliance With Llm Reviewing Policy:**

Affirmed.

**Key Questions For Authors:**

- Please precisely specify which parameters are aggregated across clients and which remain per-expert at the server.
- The number of experts N is small. How does FedWave behave when there are many clients per role (true FL scale)? Can the method support that scenario?
- It is suggested to provide an ablation study isolating the contribution of each VCL loss (pos, cont, cons) and each router regularizer (load balancing or entropy).

**Limitations:**

Yes

**Strengths And Weaknesses:**

Strengths:
-The problem of preserving both data locality and workflow dependencies is interesting and important, providing good motivation for FedWave.
-The validation is comprehensive, with extensive experiments on three domains, three backbones, multiple FL optimizers, and comparisons to centralized MAS baselines.
-The VCL losses and DPO objective are clearly defined.

Weaknesses:
-It is unclear how stage-specific modules are handled server-side: which parameters are aggregated across clients and which are kept as per-expert components, e.g., stage-specific VCL and LoRA? In addition, the authors alternately use global and individual weights, but the aggregation/update protocol per module is not fully explained.
-The setting effectively uses a small number of experts (N=4) with full participation each round. This seems like multi-task co-training across several roles rather than typical cross-silo FL, limiting FedWave’s generality.
-Ablations do not examine the VCL losses individually, nor do they compare the effect of load balancing and entropy regularization in the router. It is unclear which inductive biases matter most.
-Some typos should be corrected, say ‘the data that make such collaboration effective”.

---

> ### Author Rebuttal · Authors · 2026-03-31
>
> Thank you for your thoughtful review and constructive comments.
>
> ---
>
> **W1 / Q1. Clarification of stage-specific modules and the aggregation/update protocol**
>
> In FedWave, the trainable state consists of two parts: **shared global components** and **expert-specific components**. These two parts are updated differently.
>
> More concretely:
> - $W_g^t$: the **shared global trainable state** at round $t$
> - $W_i^t$: the **trainable state associated with expert $i$** at round $t$
>
> The update protocol is:
>
> | Module | Scope | Server-side handling |
> |---|---|---|
> | MoE router $W_{\mathrm{router}}$ | Shared global module | **Updated locally on clients and aggregated globally on the server with standard FL aggregation** |
> | LoRA adapter of expert $i$ | Expert-specific trainable module | **Updated locally for expert $i$, and aggregated only within the same expert slot; not averaged across different workflow stages** |
> | VCLayer of expert $i$, $\{W_{qkv,i}, W_{o,i}\}$ | Expert-specific workflow module | **Updated locally for expert $i$, and maintained or aggregated only within the same expert slot; not mixed across different workflow stages** |
> | Frozen base LLM | Shared backbone initialization | Frozen, not updated |
>
> Thus, when we refer to global weights $W_g^t$, we mean the **shared state maintained at the server**, most importantly the router-related component. When we refer to local weights $W_i^t$, we mean the **trainable parameter block of expert $i$**, including its LoRA adapter and VCLayer. These expert-specific modules are kept separate across workflow stages. If multiple clients correspond to the same expert slot, aggregation is performed only within that slot and never across different stages.
>
> ---
>
> **W2 / Q2. Small number of experts and limited cross-silo generality**
>
> To directly address this concern, we conducted an additional experiment where the number of workflow roles is fixed at **4**, while each role is further split into multiple federated clients:
>
> | Setting | Total clients | Participating clients / round | METEOR |
> |---|---:|---:|---:|
> | 1 client per role (original) | 4 | 4 | **24.83** |
> | 2 clients per role | 8 | 4 | **25.31** |
> | 4 clients per role | 16 | 4 | **25.07** |
> | 8 clients per role | 32 | 4 | **25.28** |
>
> These results show that **FedWave remains stable when scaling from one client per role to many clients per role**, even when the number of participating clients per round is fixed. In addition, the original paper already reports an expert-scaling analysis up to **32 experts in Figure 3**. The two analyses are complementary: **Figure 3 studies scaling in the number of workflow experts**, whereas the new experiment studies scaling in the number of **clients per role**. Together, they provide stronger evidence that FedWave is not restricted to the smallest 4-expert configuration.
>
> ---
>
> **W3 / Q3. Isolating the contribution of each VCL loss and each router regularizer**
>
> We have now conducted a finer-grained ablation study to isolate the contribution of each VCL term and each router regularizer:
>
> | Variant | METEOR |
> |---|---:|
> | Full FedWave | **24.83** |
> | w/o $\mathcal{L}\_{pos}$ | 24.41|
> | w/o $\mathcal{L}\_{cont}$ | 24.30 |
> | w/o $\mathcal{L}\_{cons}$ | 24.28 |
> | w/o $\mathcal{L}\_{balance}$ | 24.32 |
> | w/o $\mathcal{L}\_{entropy}$ | 24.57 |
> | w/o $\mathcal{L}\_{balance} + \mathcal{L}\_{entropy}$ | 24.17 |
>
> The results show that **all three VCL terms contribute positively** to the final performance. In this setting, removing $\mathcal{L}\_{cont}$ or $\mathcal{L}\_{cons}$ causes the largest drop, while $\mathcal{L}\_{pos}$ also provides a clear gain. We also find that **both router regularizers are beneficial**, and removing both leads to the largest degradation among the router variants, indicating that $\mathcal{L}\_{balance}$ and $\mathcal{L}\_{entropy}$ are **complementary rather than redundant**. We will add this fine-grained ablation table to the revision and discuss more explicitly which inductive biases matter most.
>
> ---
>
> **W4. Typos and minor wording issues**
>
> Thank you for catching this. We will carefully proofread the manuscript and correct this typo as well as other minor wording issues.

---

> > ### Author Rebuttal · Reviewer_NapB · 2026-04-03
> >
> > The rebuttal solved my concerns, so I maintain my score.

---

### Decision · Program_Chairs · 2026-04-30

**Decision:**

Accept (regular)

**Comment:**

This paper presents FedWave, which enables federated and workflow-aware collaboration for distributed, privacy-preserving sequential workflow scenarios. Reviewers acknowledge the importance of privacy preservation for LLM inference. However, they raise several critical concerns regarding the clarity of presentation and the scalability of the experimental setup, including server-side parameters, overstated contributions, an insufficient number of experts, and limited datasets used for evaluation. Although the authors provided detailed responses during the rebuttal, several concerns remain unresolved. Overall, the paper requires substantial revisions, such as better positioning its contributions, including more datasets for evaluation, and increasing the number of experts. Without the revisions suggested by reviewers, the current quality is insufficient for publication at ICML.